# Material composition and constitutive model development of red mud-based filler for highway tunnel invert filling applications: A comprehensive study

Xiaolong Song[1,2], Junying Rao[1,2]*, Hongchao Cui[3], Mingwei He[4]

**1** Research Center of Space Structures, Guizhou University, Guiyang, Guizhou, China, **2** Guizhou Province Key Laboratory of Green Building and Intelligent Construction, Guiyang, Guizhou, China, **3** Datun Coal and Electricity Co., Ltd., Xuzhou, Jiangsu, China, **4** School of Civil Engineering, Guizhou Institute of Technology, Guiyang, Guizhou, China

* jyrao@gzu.edu.cn

## Abstract

To address the global environmental challenges posed by red mud waste, this paper proposes utilizing red mud as a filling material for highway tunnel inverts. It is imperative to thoroughly investigate the theoretical foundation of red mud-based filling (RMBF) materials for this application. This study explores the optimal ratio of RMBF and, drawing on the results of triaxial compression tests in conjunction with soil structural parameters and Duncan-Chang theory, proposes and defines for the first time the strengthening parameter ($S_p$) of RMBF to derive its constitutive model. The test data were systematically compared and analyzed with the results of theoretical calculations. The findings indicate that RMBF, which is composed of red mud, cement, steel slag, slaked lime, EDTA-2Na, water and curing agent mixed in a specific proportion, can meet the strength requirements. When used for filling the invert of a highway tunnel, 1.068 tons of red mud can be recycled from 1 cubic meter of RMBF. The stress-strain curve of the RMBF exhibits an evident peak, indicating strain softening characteristics, while the strengthening stress-strain curve clearly demonstrates strain hardening characteristics. The discrepancy between the stress-strain curve predicted by the RMBF model considering $S_p$ and the actual test curve is within 10%, confirming the model's rationality and applicability.

## 1. Introduction

The accumulation and disposal of red mud pose serious environmental and safety challenges around the world [1] Red mud is an insoluble industrial waste produced as a byproduct during the extraction of alumina from bauxite. For every ton of alumina produced, about 1.5 tons of red mud are generated [2]. With the rapid development of the aluminum industry, the demand for alumina is increasing, and a large amount of red mud is discharged. As a major producer of alumina, China discharges millions of tons of red mud every year [3]. Due to the high heavy metal content and strong alkalinity of red mud, the current main method of storing it in open-air dams on land is not only expensive to maintain but also prone to safety

**Data availability statement:** All relevant data are within the manuscript and its Supporting Information files.

**Funding:** This research was sponsored by the Guizhou Provincial Science and Technology (Grant No. ZK[2023]80 and No. ZK[2021]284). The funders had no role in study design, data collection and analysis, decision to publish, or preparation of the manuscript.

**Competing interests:** The authors have declared that no competing interests exist.

accidents. In recent years, red mud leakage incidents have been reported in Hungary [4], Vietnam, China, and India. These incidents have resulted in serious pollution of water bodies and farmland, and even caused casualties. Although the global average utilization rate of red mud has reached 15% [5], China's comprehensive utilization rate of red mud remains at only 5.69% (Fig 1), far below the world average. Faced with the significant challenges of red mud accumulation and utilization, China urgently needs to explore more effective strategies for resource utilization.

The most effective approach to recycling red mud lies in the field of engineering materials [6]. Many researchers have studied the potential of red mud for a broad spectrum of applications, including its feasibility as a raw material for concrete, mortar, building blocks, tiles, road subgrade, and more [7–17]. Due to the high alkalinity and heavy metal contamination of red mud, it must undergo modification. We have compiled commonly used modification materials from related studies in both Chinese and international literature, primarily including cement, fly ash, slag, metal slag, lime, and others (Table 1). The results indicate that adding these materials improves the strength and other mechanical properties of the mixture to varying extents. Notably, Chen et al. [18] suggested using red mud to replace part of the loess in roadbed construction. Through a series of experiments, it was observed that when the red mud content was 15%–20%, the mechanical properties of the loess roadbed were optimal, exhibiting high compressive strength and good dynamic properties. Similarly, based on previous research conducted by our group [19], we have proposed a novel utilization approach: modifying red mud for use as a filling material for highway tunnel inverts (Fig 2). Concerning tunnel invert filling materials, the following regulations are applicable [20]: For tunnels with inverts, the invert filling layer can serve as the roadbed, and the strength grade of the invert filling material should not be lower than C15 concrete or rubble concrete.

To ensure reliability in engineering applications, it is essential to establish the corresponding constitutive model. In establishing soil constitutive relations, the common approach involves combining indoor tests, theoretical calculations, and numerical simulations. Yan et al. [21] applied the Ramberg-Osgood strain hardening law to the fatigue

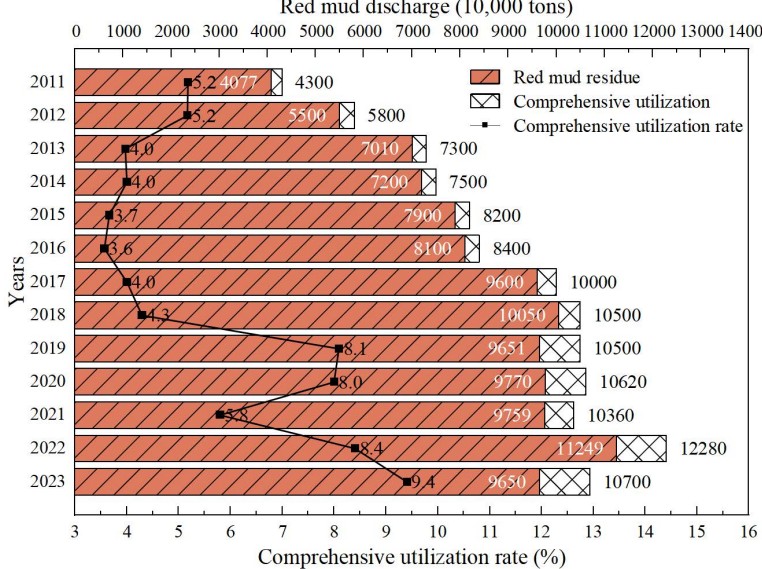

**Fig 1. Red mud emissions and comprehensive utilization rate in China in the past decade (incomplete statistics).**

damage analysis of mud interlayers using extensive dynamic triaxial tests and defined a new fatigue damage variable expression. Ji et al. [22] combined the Lemaitre strain equivalence hypothesis and Weibull strength theory and applied these principles to constructing a damage model for waste brick concrete for the first time. The constitutive model parameters were obtained from uniaxial compression test results. Phan et al. [23] studied the transition from diffuse failure to localized failure in partially saturated soil under stress conditions and introduced parameters such as the shear band's width and inclination angle. They proposed a thermodynamics-based constitutive model that accurately describes the stress-strain behavior and size effects in different regions of the soil post-localized failure. The model's effectiveness was validated through numerical simulation. Chang et al. [24] examined the effects of particle crushing and plastic shear on the mechanical properties of frozen salt coarse sand during loading and proposed a constitutive model combining particle crushing and plastic shear mechanisms. Experimental validation demonstrated that the model accurately predicts the stress-strain behavior of frozen salt coarse sand under varying salt contents and stress conditions. Liu et al. [25] developed a constitutive model describing the

**Table 1. Statistics of red mud modified materials.**

| Materials | Literatures |
| --- | --- |
| Cement | [6,18,34–42] |
| Fly ash | [6,39,42–47] |
| Slag | [34,48–51] |
| Metal slag | [6,48,49,51–53] |
| Natural products | [40,45,54–56] |
| Lime | [6,34,39,57,58] |
| Desulfurization gypsum | [44,51,57,58] |
| Coal Gangue | [43,52] |
| Polymer stabilizer | [34,35] |
| Metakaolin | [38] |
| Rubber powder | [36] |
| Waste concrete | [59] |

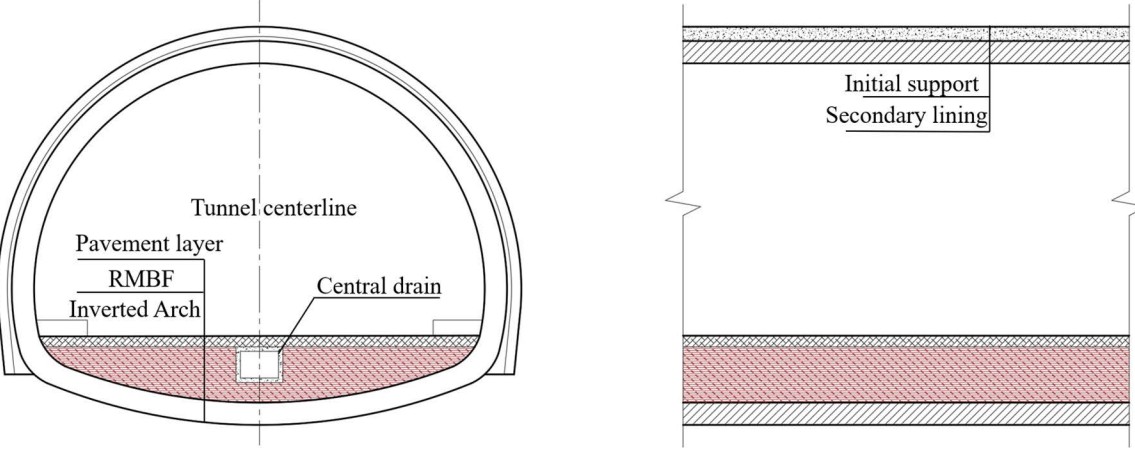

**Fig 2. Schematic diagram of RMBF backfill inverted arch.**

mechanical behavior of tailings soil under freeze-thaw cycles based on mesomechanics and homogenization theory. The model accurately captures the stress-strain behavior of tailings soil under different confining pressures and freeze-thaw cycles and accounts for strain hardening and post-peak strain softening phenomena.

At present, the models describing the constitutive relationship of soil can be broadly categorized into two types: elastic models and elastoplastic models. Elastic models encompass both linear and nonlinear elastic models, such as the *K-G* model and the *E-μ* model. Elastoplastic models comprise the Cambridge model [26], the Wright-Duncan model [27], the Tsinghua model [28], and the double yield surface model [29], among others. In theory, the elastoplastic model can more accurately simulate the complex mechanical behavior of soil; however, due to its complex computational process and the difficulty in obtaining parameters, its application in practical engineering is limited. Conversely, the nonlinear elastic model is more widely used in engineering because its parameters are easier to obtain, the computational process is relatively simple, and it can better simulate the actual mechanical properties of soil. The Duncan-Chang model [30] is a representative *E-μ* model. Cai et al. [31] carried out a functional zoning optimization design for a cemented gravel dam based on an improved Duncan-Chang nonlinear elastic model to reduce material consumption and ensure dam safety. Under uniaxial compression conditions, Liu et al. [32] derived the mechanical properties of freeze-thaw modified sand and proposed a nonlinear elastic strain hardening model by combining the Duncan-Chang model with the elastic strain hardening model to accurately describe its stress-strain behavior. Liu et al. [33] investigated the effects of freeze-thaw cycles on the mechanical properties of lean clay and proposed a new model, which, by modifying the parameters of the Duncan-Chang model, is more suitable for civil engineering design in cold regions.

Scholars have conducted limited research on highway tunnel invert filling materials, with even fewer studies on their constitutive relationships. Building on the prior research conducted by our group, this study determined the optimal ratio of red mud-based fillers (RMBF) through compaction tests and unconfined compressive strength tests. Through triaxial compression tests, the stress-strain curve of the RMBF was obtained, and the post-compression failure morphology of the RMBF was observed. Referring to the structural parameters of soil, the strengthening parameter ($S_p$) was proposed and defined. Following the Duncan-Chang theory, a constitutive model of the RMBF incorporating the strengthening parameter was established. Finally, the model was verified through a comparative analysis of experimental data and theoretical calculations. This study offers a theoretical basis for the application of red mud waste in highway tunnel invert filling, which presents significant environmental and economic benefits.

## 2. Test materials and test methods

### 2.1. Test materials

The red mud used in this experiment was taken from the tailings pond in Qingzhen City, Guiyang, and stored in an open-air storage method. The red mud taken on site is naturally bonded and solidified, and the particles are relatively large. In order to ensure that it can be fully mixed during sample preparation, the red mud is crushed with a wooden mallet and passed through a 2 mm sieve for sample preparation. The research team found in its previous work that the composite material prepared with cement, steel slag, slaked lime, and EDTA-2Na in a mass ratio of 1:2:1:1 has the best effect on enhancing the performance of red mud [19]. The material sample is shown in Fig 3.

## 2.2. Compaction test and unconfined compressive strength test

The compaction test was conducted following the Highway Geotechnical Test Specification (JTG 3430–2020) [60] to determine the maximum dry density and the optimum moisture content of the mixture. The mixture was prepared by the dry soil method, beginning with a moisture content of 18%, increasing by 2% increments. The mixture was thoroughly mixed and then tested after being sealed for 24 hours. The experiment consisted of 7 groups of tests, with a total of 35 specimens (Table 2). The test utilized the heavy II-2 compaction method, involving compaction in three layers.

The UCS test was conducted following the Test Specification for Stabilized Materials with Inorganic Binders for Highway Engineering (JTG E51-2009) [61]. The specimens were prepared based on the optimum moisture content. To account for compaction effects during construction, three compaction degrees of 90%, 93%, and 96% were selected. The specimens were formed using hydraulic molds and placed in a standard curing box (temperature 20±2°C, humidity above 95%) for 7 days and 28 days. Three specimens were prepared in each group, resulting in a total of 126 specimens (Table 3). The test instrument used was a YYW-11 unconfined pressure instrument, manufactured by Tianjin Gangyuan Testing Instrument Factory, with a loading rate of 3 mm/min.

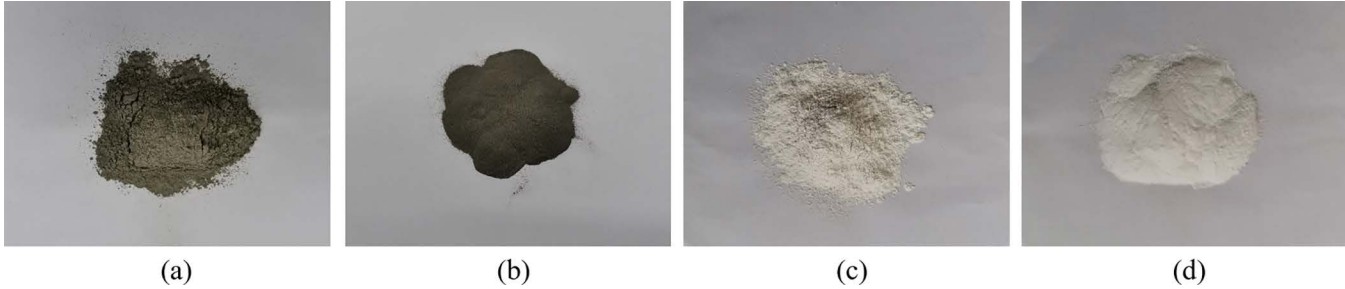

| (a) | (b) | (c) | (d) |

**Fig 3. Sample collection and preparation.** (a) Cement, (b) Steel slag, (c) Slaked lime, (d) EDTA-2Na.

**Table 2. Compaction test plan.**

| Composite modified materials/% | 0 | 5 | 10 | 15 | 20 | 25 | 30 |
|---|---|---|---|---|---|---|---|
| Maintenance age/d | | | 18~26 | | 14~22 | | 12~20 |
| Number of test pieces | | | 20 | | 10 | | 5 |
| Total | | | 35 | | | | |

**Table 3. Unconfined compressive strength test plan.**

| Composite modified materials/% | 0 | 5 | 10 | 15 | 20 | 25 | 30 |
|---|---|---|---|---|---|---|---|
| Maintenance age/d | | | | 7, 28 | | | |
| Compaction/% | | | | 90, 93, 96 | | | |
| Number of test pieces | | | | 126 | | | |

## 2.3. Triaxial compression test

According to the cross section of a highway tunnel project, the bottom of the tunnel consists of a 9 cm thick asphalt concrete layer, a 24 cm thick C45 concrete layer, and a 15 cm thick C20 concrete layer from top to bottom, with RMBF used to fill the invert. The bulk densities and thicknesses of these layers are shown in Table 4. The asphalt concrete used here is of the AC-20 type, and the calculated self-weight stress transmitted from the pavement structure to the top of the red mud-based filler is 11.5 kPa.

The self-weight stress transmitted to the bottom of the invert by the pavement structure and RMBF is 46.7 kPa. Du et al. [62] found through monitoring that the stress response range of the filling layer of the invert of a Chinese tunnel is between 15 kPa and 55 kPa. Adding the self-weight stress, the confining pressure of the RMBF in this paper is set between 26.5 kPa and 101.7 kPa.

The confining pressures used in this study are 30 kPa, 60 kPa, and 90 kPa. The specimens were prepared using a three-valve membrane preparation device with the optimal ratio of RMBF. The compaction degree was 96%. After preparation, the specimens were placed in a standard curing box for curing according to the specified curing duration. Fig 4a depict the mold and specimen. The triaxial compression test plan is shown in Table 5, comprising a total of 9 groups. The GDSTTS test system, developed by the British company GDS Instruments, was used, consisting of a pressure chamber, a hydraulic control system, and a PC operating terminal (Fig 4b).

## 3. Test results

### 3.1. Optimum moisture content

After compaction, the scraped sample was weighed, and a representative sample from the center was taken to measure its moisture content. The dry density of each specimen was then calculated. The compaction test curve is depicted in Fig 5. The maximum dry density, identified

**Table 4. Self-weight stress calculation.**

| Materials | Test weight (N/m³) | Thickness (cm) | Self-weight stress (kPa) |
|---|---|---|---|
| Asphalt concrete | 24400 | 9 | 2.2 |
| C45 Concrete | 24000 | 24 | 5.8 |
| C20 Concrete | 23600 | 15 | 3.5 |
| Pavement interface | \ | \ | 11.5 |
| RMBF | 20700 | 170 | 39.2 |
| Invert interface | \ | \ | 46.7 |

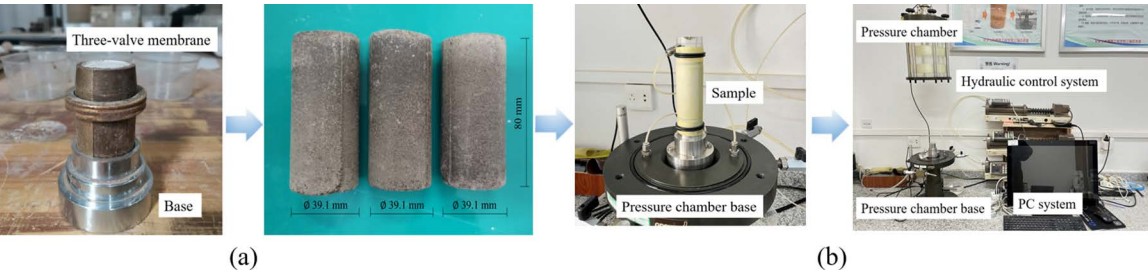

(a)                                                    (b)

**Fig 4. Triaxial test process.** (a) Sample Preparation, (b) GDSTTS test system.

as the peak point, was determined, and the moisture content corresponding to this point was considered the optimal moisture content. After adding the composite modified material, the density of the red mud-based mixture significantly improved. As the amount of the composite modified material increased, the maximum dry density of the mixture increased almost uniformly, while the optimal moisture content decreased in a similar manner. The test results are presented in Table 6.

## 3.2. Unconfined compressive strength

Fig 6 depicts the main forms of specimen failure. When the specimen exhibits approximately 90° vertical through-type cracks, it indicates that the red mud-based mixture has reached its ultimate compressive strength. The average value of $q_u$ from the three specimens is taken as the unconfined compressive strength of the group. As the amount of composite modified materials increases, the unconfined compressive strength of the red mud-based mixture gradually increases, and the effect of compaction follows the same trend. Higher compaction levels result in higher unconfined compressive strength of the mixture (Fig 7).

Table 5. Triaxial compression test plan for RMBF.

| Sample number | Age (d) | Confining Pressure (kPa) |
|---|---|---|
| TA 1~TA 3 | 7d | 30、60、90 |
| TB 1~TB 3 | 14d | 30、60、90 |
| TC 1~TC 3 | 28d | 30、60、90 |

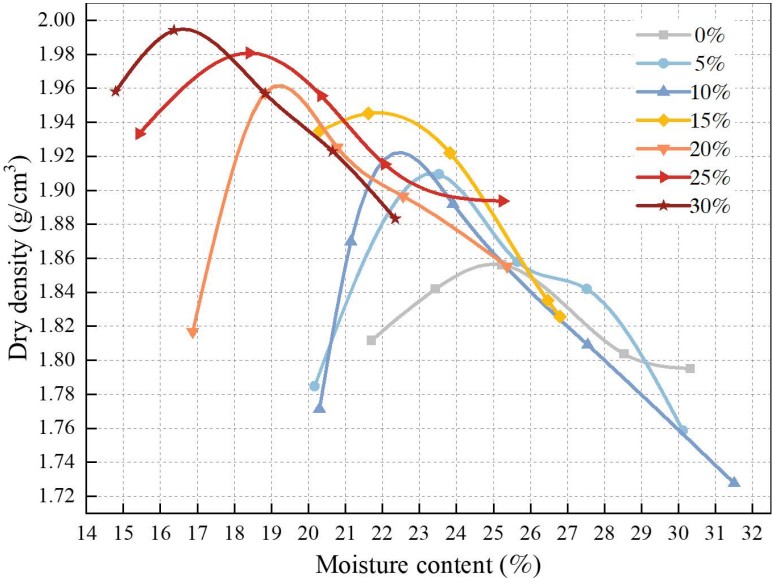

Fig 5. Compaction test curves.

Table 6. Compaction test results.

| Composite modified material dosage (%) | 0 | 5 | 10 | 15 | 20 | 25 | 30 |
|---|---|---|---|---|---|---|---|
| Maximum dry density (g/cm³) | 1.86 | 1.91 | 1.92 | 1.95 | 1.96 | 1.98 | 1.99 |
| Optimum moisture content (%) | 25.1 | 23.4 | 22.4 | 21.9 | 19.2 | 18.4 | 16.6 |

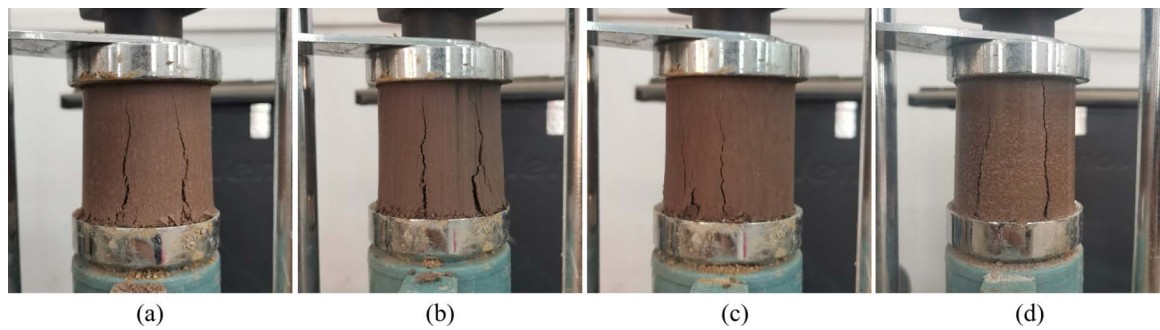

**Fig 6. Failure mode of UCS test for MRM (28d, 96% compaction degree).** (a) 5%, (b) 15%, (c) 20%, (d) 30%.

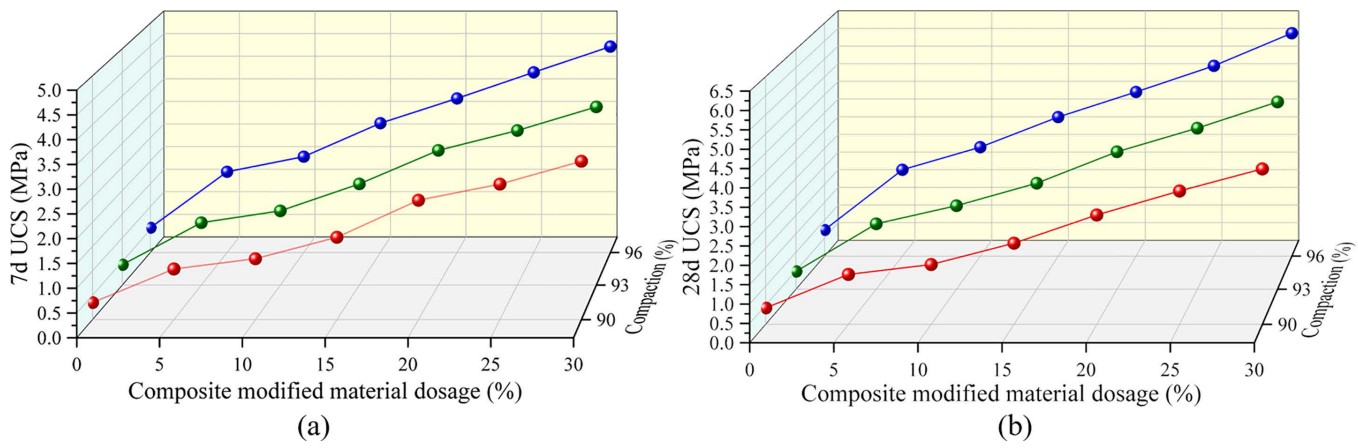

**Fig 7. Unconfined compressive strength of MRM.** (a) 7d, (b) 28d.

As shown in Fig 8, the unconfined compressive strength of the modified red mud (MRM) shows a significant increase. For specimens at a curing age of 7 days, the strength increase ranged from 197% to 226%, with an average increase of 210% when the composite modifier was initially added. As the amount of composite modifier continued to increase, the strength gain tended to stabilize, with an increase of 14% to 45%, and an average increase of 23.5%, showing a gradually decreasing trend. For specimens at a curing age of 28 days, the trend was consistent with that of the 7-day specimens. When the composite modifier content was 5%, the average increase reached 233%. As the content continued to increase, the average increase decreased to 23.4%.

### 3.3. Proportion of RMBF

At a composite modified material content of 30% and a compaction degree of 96%, the 28d unconfined compressive strength of MRM reaches 6.22 MPa, which is still lower than the 7.2 MPa compressive strength requirement for invert filling in the specification [20]. To address this, soil stabilizers were introduced, with resin polymer as the main component. In this study, two types of stabilizers, Tushengda (TSD) and Suzhou Luxing (SZLX), were used to mix with MRM containing 30% composite modified materials. As shown in Fig 9, with the addition of 5-fold diluted TSD stabilizer, the 28d unconfined compressive strength of MRM reaches 7.37 MPa, which meets the requirements of the specification.

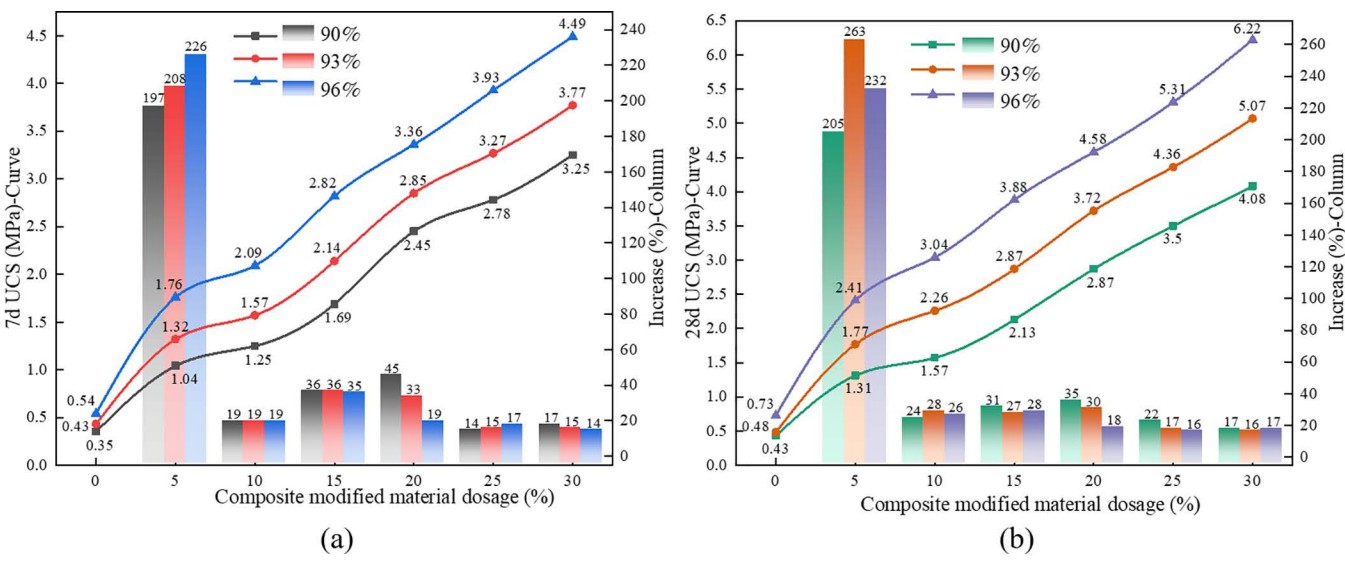

**Fig 8. The changing trend of unconfined compressive strength of MRM.** (a) 7d, (b) 28d.

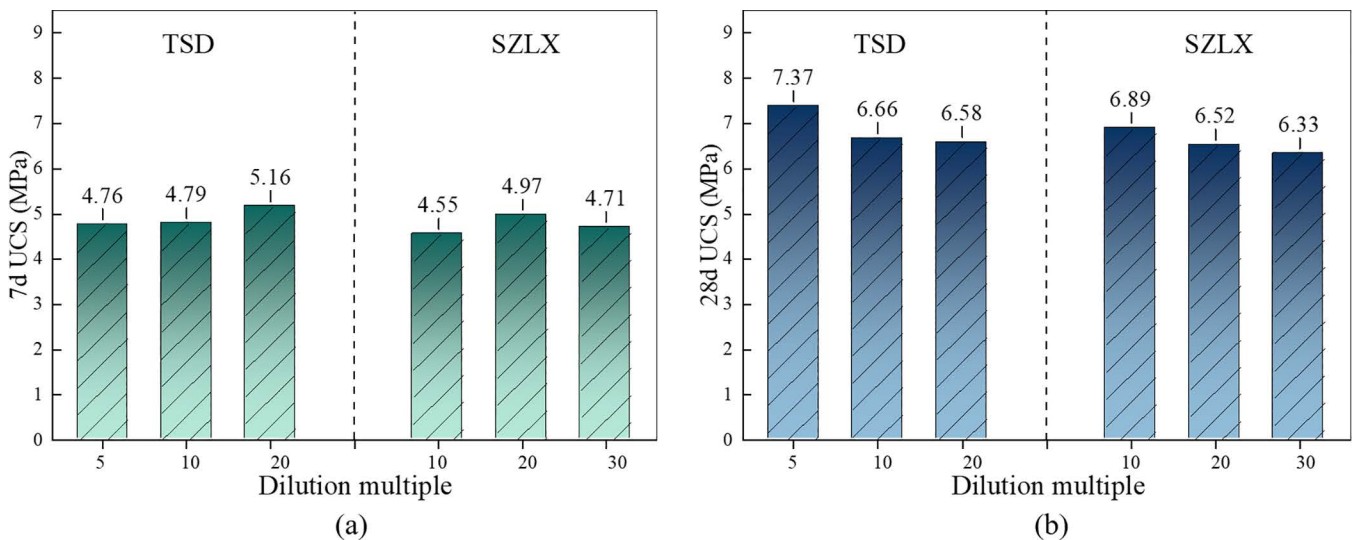

**Fig 9. Unconfined compressive strength of MRM after adding soil stabilizers.** (a) 7d, (b) 28d.

The composition of RMBF has been determined as 51.7% red mud, 29% composite modified material, 16.1% water, and 3.2% 5-fold diluted TSD curing agent. The current density of RMBF is 2.07 t/m³, allowing 1 m³ of RMBF to recycle 1.068 t of red mud. Additionally, when used for highway tunnel invert filling, the compaction degree of RMBF should be no less than 96%.

## 3.4. Stress-strain curve

As shown in Fig 10, the stress-strain curve of RMBF exhibits distinct peak points, indicative of strain softening characteristics. Taking the 28d-30kPa test conditions as an example (Fig 10d),

the deformation and failure process of RMBF under triaxial compression can be divided into five stages:

Stage 1: Compaction: The stable stage at the beginning of the curve. Under the action of confining pressure and axial stress, the voids within RMBF are compacted, resulting in an increase in axial strain without significant stress change. The strain during the compaction stage reaches 2.95%~3.65% under 30 kPa and 60 kPa confining pressures, and 0.73%~1.77% under 90 kPa confining pressure.

Stage 2: Elastic Deformation: The curve rises rapidly to the stage where fluctuations occur. During this stage, RMBF stores elastic energy and returns to its original state after unloading, characterizing elastic deformation.

Stage 3: Plastic Deformation: The curve fluctuates to the peak point. Following the elastic stage, fluctuations in stress occur. At this point, RMBF has reached its energy storage limit; microcracks appear inside, energy is dissipated, stress is released, and then it reaches the peak value.

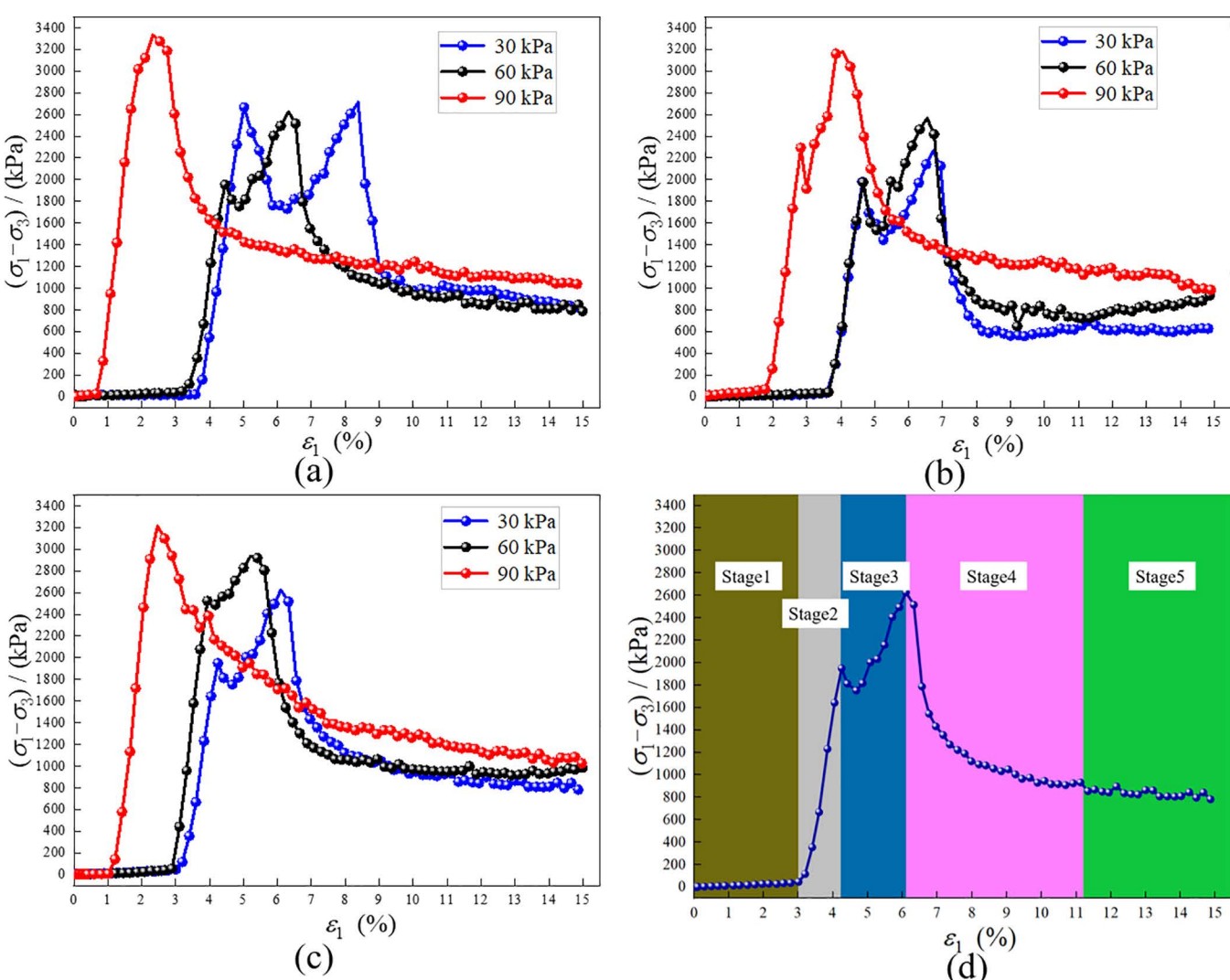

**Fig 10. The ($\sigma_1$-$\sigma_3$)-$\varepsilon_1$ curves of RMBF with confining pressure under different ages.** (a) 7d, (b) 14d, (c) 28d, (d) 28d-30kPa.

Stage 4: Yield: The stage of rapid decline after the peak point of the curve. As the internal microcracks develop and penetrate, RMBF reaches its strength limit; the curve drops rapidly after the peak, the strain increases quickly, and the specimen fails.

Stage 5: Post-Peak Stability: The final stable stage of the curve. Due to the confining pressure, RMBF still retains a certain strength after yielding, and the stress tends to stabilize. However, the strain continues to increase rapidly, exhibiting shear dilatancy. The sample undergoes shear failure, a large gap appears at the shear surface, and the sample expands outward (Fig 11).

As shown in Fig 12, the elastic modulus and peak stress of RMBF exhibit consistent trends, but no obvious linear relationship with curing age is observed. However, as confining pressure increases, both elastic modulus and peak stress also increase, indicating the influence of confining pressure on the soil's compressive strength and its impact on RMBF strength.

## 4. Constitutive model analysis

Eq. (1) is the $E$-$\mu$ constitutive equation within the Duncan-Chang model. The eight material constants required for calculation are: $K$, $n$, $R_f$, $c$, $\varphi$, $D$, $G$, and $F$. Among them, $K$, $n$, $R_f$, $c$, and $\varphi$ are used to calculate the tangent Elastic modulus of the material; $D$, $G$, and $F$ are used in calculating the tangent Poisson's ratio of the material, and when combined with the first five constants, they determine $\mu_t$. The derivation process indicates that the material constants of the Duncan-Chang model can all be obtained through triaxial compression tests.

$$
\left\{
\begin{array}{l}
E_t = K \cdot p_a \cdot \left(\dfrac{\sigma_3}{p_a}\right)^n \cdot \left[1 - R_f \cdot \dfrac{(\sigma_1 - \sigma_3) \cdot (1 - \sin\varphi)}{2 \cdot c \cdot \cos\varphi + 2 \cdot \sigma_3 \cdot \sin\varphi}\right]^2 \\[4mm]
\mu_t = \dfrac{G - F \cdot \lg(\sigma_3 / p_a)}{\left\{1 - \dfrac{D \cdot (\sigma_1 - \sigma_3)}{K \cdot p_a \cdot \left(\dfrac{\sigma_3}{p_a}\right)^n \cdot \left[1 - R_f \cdot \dfrac{(1 - \sin\varphi) \cdot (\sigma_1 - \sigma_3)}{2 \cdot c \cdot \cos\varphi + 2 \cdot \sigma_3 \cdot \sin\varphi}\right]}\right\}^2}
\end{array}
\right.
\tag{1}
$$

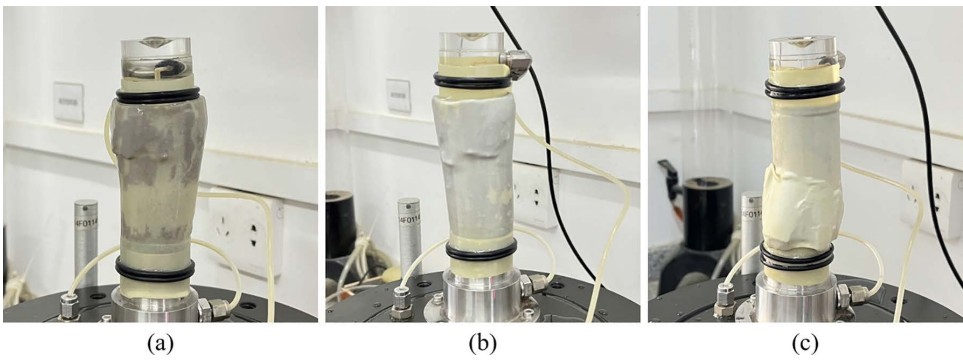

(a) (b) (c)

**Fig 11. The failure mode of triaxial compression test for RMBF.** (a) TA3, (b) TB1, (c) TC2.

## 4.1. Strengthening parameter

The triaxial compression test curve of RMBF exhibits pronounced strain softening characteristics, though the strain softening process is inherently unstable, making it challenging to establish a constitutive relationship model [63]. Therefore, this paper introduces $S_p$ to transform the strain softening curve of RMBF into a strain hardening curve. Mechanical properties such as strength, permeability, and deformation, influenced by the composition, spatial arrangement, and inter-particle interaction of the soil, are referred to as soil structure. The strength of the structure reflects the extent to which soil structure influences its mechanical properties, a concept known as the structural potential of the soil. Xie et al. [64] proposed a method to release the structural potential grounded in soil mechanics research methodologies. On this basis, it is suggested that the structural quantitative parameters of a specific soil can be derived from tests on its undisturbed, disturbed, and waterlogged states. For triaxial compression tests, Luo et al. [65] proposed that the structural parameters derived from triaxial compression tests be defined as shown in Eq. (2).

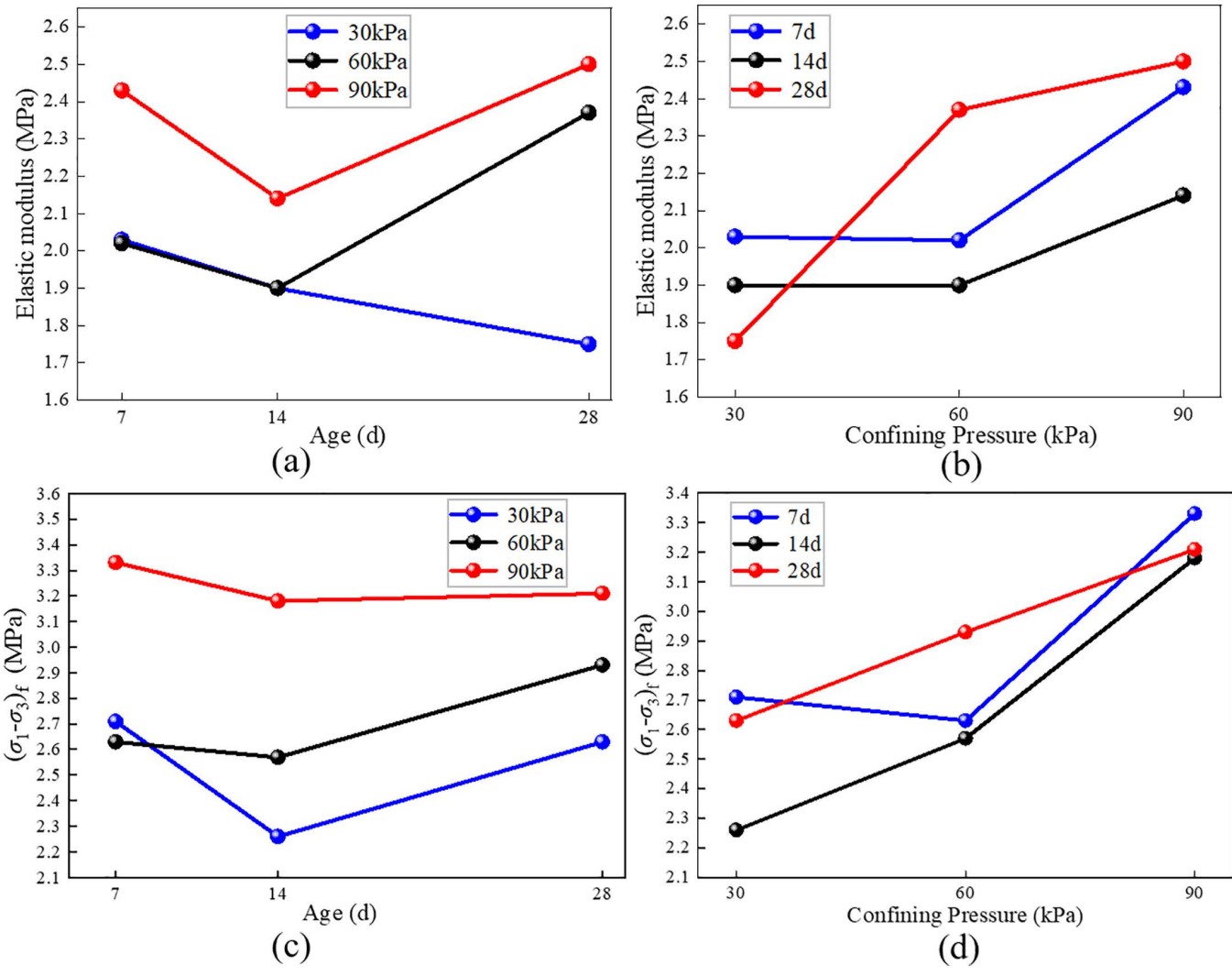

**Fig 12. The changing trend of $E$ and $(\sigma_1\text{-}\sigma_3)_f$ of RMBF.** (a) $E$ -age, (b) $E$ -confining pressure, (c) $(\sigma_1\text{-}\sigma_3)_f$ -age, (d) $(\sigma_1\text{-}\sigma_3)_f$ -confining pressure.

$$\begin{cases} m_1 = \dfrac{q_\mathrm{o}}{q_\mathrm{s}} \\[2mm] m_2 = \dfrac{q_\mathrm{r}}{q_\mathrm{o}} \\[2mm] m_\varepsilon = \dfrac{m_1}{m_2} = \dfrac{q_\mathrm{o}^{\,2}}{q_\mathrm{s} \cdot q_\mathrm{r}} \end{cases} \tag{2}$$

Among them, $q_o$, $q_s$, and $q_r$ represent the shear strengths of the original soil, water-saturated soil, and disturbed soil under the same axial strain, respectively; $m_1$ reflects the strength change due to the disruption of the arrangement characteristics of damaged soil particles, and $m_2$ reflects the strength change due to the disruption of the connection strength of damaged soil particles. The combination of these two parameters can reflect the structural properties of the soil under load-induced damage.

Through the above analysis, RMBF is considered as "original soil," undisturbed red mud (URM) as "water-saturated soil," and modified red mud (MRM) with 5% composite modified materials as "disturbed soil" to construct the structural parameters of RMBF. Specimens for the triaxial compression test were prepared based on the optimal moisture content and maximum dry density determined by the compaction test. The optimal moisture content of URM was 25.1%, with a maximum dry density of 1.86 g/cm³ and a specimen mass of 171.52 g. For MRM, the optimal moisture content was 23.4%, with a maximum dry density of 1.91 g/cm³ and a specimen mass of 176.03 g. The test conditions were consistent with those outlined in Table 3. Following the test, stress-strain curves for URM and MRM were obtained, showing changes in confining pressure over different curing ages, as illustrated in Figs 13 and 14. The test results exhibit characteristics of strain hardening, indicating that no distinct peak point was present in the curve.

Building on the definition of structural parameters, the strengthening parameter for RMBF is introduced, as shown in Eq. (3). The $S_p$ is conceptually opposite to the structural parameter and reflects the strength improvement RMBF contributes to the structural enhancement of URM.

$$S_p = \left. \dfrac{(\sigma_1 - \sigma_3)_\mathrm{r}}{(\sigma_1 - \sigma_3)_\mathrm{u}} \middle/ \dfrac{(\sigma_1 - \sigma_3)_\mathrm{m}}{(\sigma_1 - \sigma_3)_\mathrm{r}} \right. = \dfrac{(\sigma_1 - \sigma_3)_\mathrm{r}^{\,2}}{(\sigma_1 - \sigma_3)_\mathrm{u} \cdot (\sigma_1 - \sigma_3)_\mathrm{m}} \tag{3}$$

In Eq. (3), $S_p$ is the strengthening parameter, which is dimensionless; $(\sigma_1\text{-}\sigma_3)_\mathrm{r}$, $(\sigma_1\text{-}\sigma_3)_\mathrm{u}$ and $(\sigma_1\text{-}\sigma_3)_\mathrm{m}$ represent the stresses of RMBF, URM, and MRM under the same axial strain, with units in kPa.

Since the compaction stage involves the closing of pores inside the sample, during which the sample does not bear strength, this stage is excluded from the calculation of strengthening parameter. The strengthening parameter are calculated based on the stresses of RMBF, URM, and MRM under the same strain; however, as the strain values of each sample differ, nonlinear polynomial fitting is performed using Matlab on the stress-strain curves of URM and MRM. As shown in Figs 13 and 14, the fitting curve closely approximates the actual curve. Within the definition domain, the fitting function provides an accurate calculation of stress. Based on the strain value of RMBF, the stress values of URM and MRM under the same strain as RMBF can be derived by substituting into the fitting function. The $S_p$ of RMBF can then be calculated using Eq. (3).

As shown in Fig 15, the strengthening parameter remains consistently greater than 1. When $ds_p/d\varepsilon_1 > 0$, it signifies that RMBF contributes positively to bearing strength; when $ds_p/d\varepsilon_1 \leq 0$, it signifies that RMBF negatively impacts bearing strength and the specimen is damaged. The trend of RMBF's strengthening parameter with strain mirrors that of stress, displaying distinct peak points and notable fluctuations. When the peak point appears, the strain of RMBF ranges between 1% and 2%. The strengthening parameter declines rapidly after the peak point is reached. When the strain reaches 5%, the strengthening parameter of RMBF approaches 1, indicating that RMBF has yielded and its strength is diminished.

## 4.2. Strengthening stress-strain curve

As shown in Fig 15, after excluding the compaction stage, the trend of the strengthening parameter with strain closely resembles the trend of stress with strain as shown in Fig 10. Consequently, the $(\sigma_1-\sigma_3)_r/s_p - \varepsilon_1$ curve is plotted, as illustrated in Fig 16. The adjusted stress-strain curve of RMBF resembles a hyperbolic type, akin to the stress-strain curve of general soil

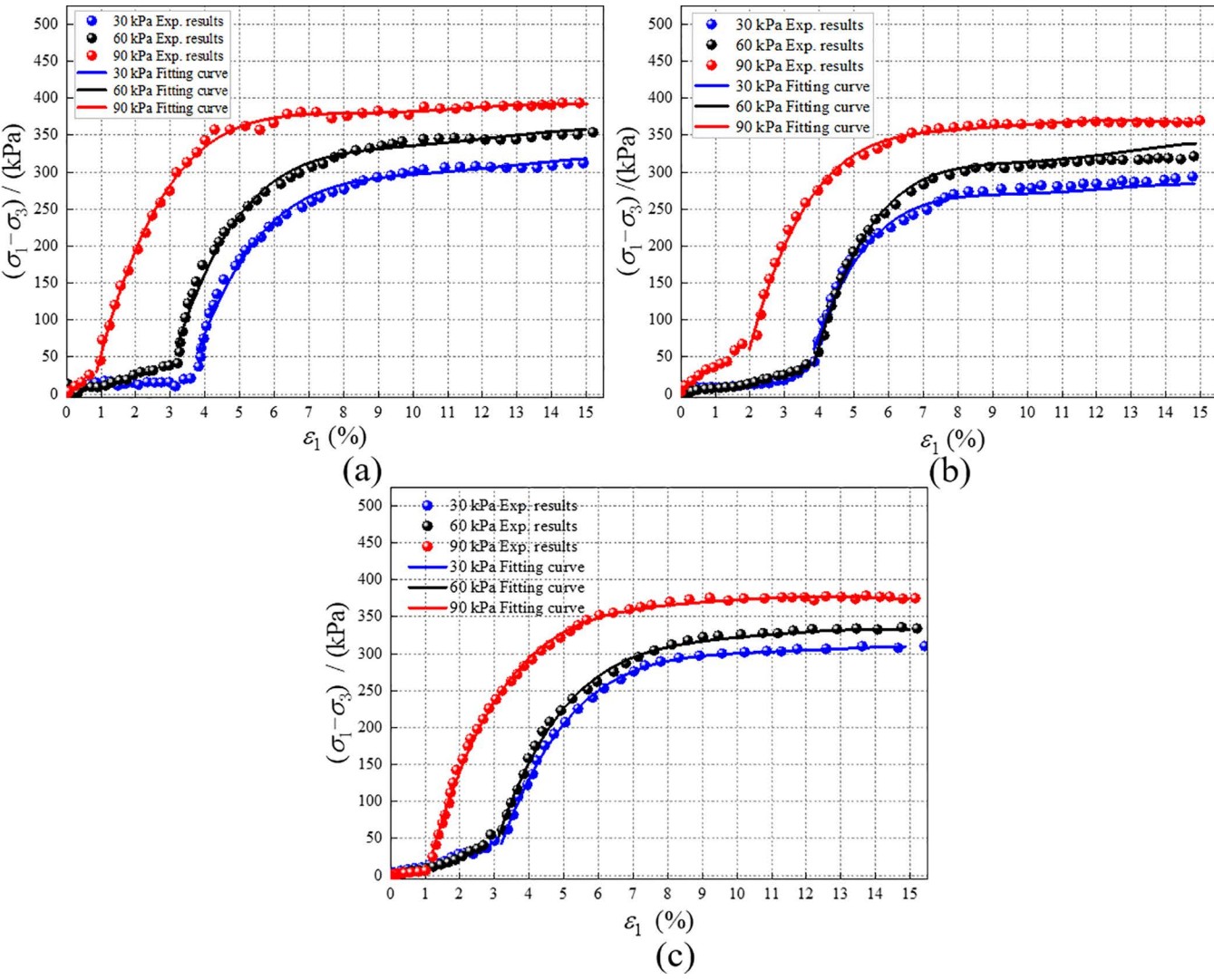

**Fig 13. The $(\sigma_1-\sigma_3)$-$\varepsilon_1$ curves of URM with confining pressure under different ages.** (a) 7d, (b) 14d, (c) 28d.

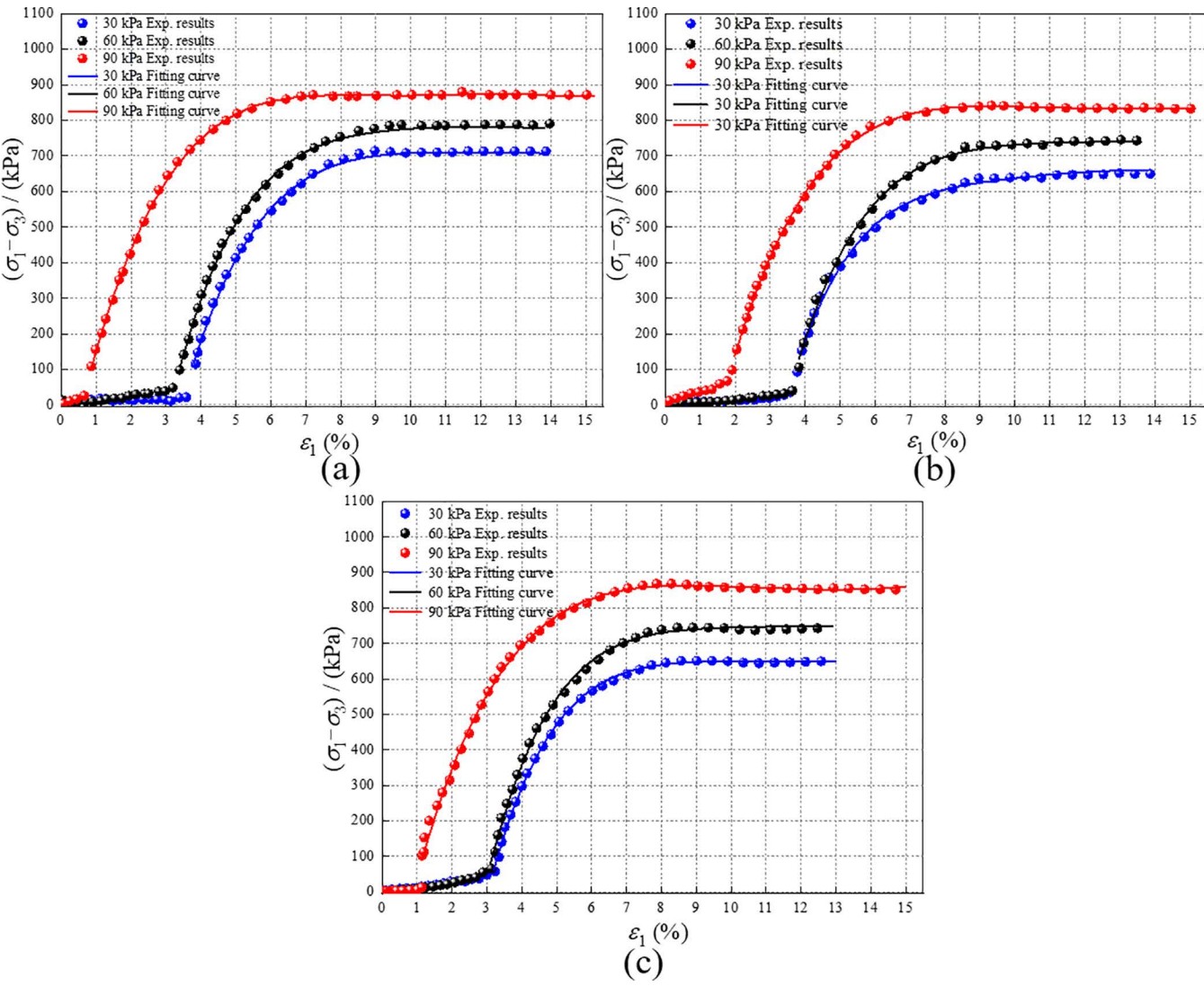

**Fig 14.  The ($\sigma_1$-$\sigma_3$)-$\varepsilon_1$ curves of MRM with confining pressure under different ages.** (a) 7d, (b) 14d, (c) 28d.

[66]. ($\sigma_1$-$\sigma_3$)$_r$/$s_p$ is defined as the strengthening stress ($\sigma_1$-$\sigma_3$)$_s$ of RMBF, as presented in Eq. (4). Subsequently, the strengthening stress-strain relationship of RMBF can be calculated using the Duncan-Chang model.

$$(\sigma_1 - \sigma_3)_s = \frac{(\sigma_1 - \sigma_3)_r}{s_p} \tag{4}$$

In Eq. (4), ($\sigma_1$-$\sigma_3$)$_s$ represents the strengthening stress of RMBF, the unit is kPa; ($\sigma_1$-$\sigma_3$)$_r$ represents the stress of RMBF triaxial compression test, the unit is kPa.

### 4.3.  Material constants

Through the above analysis, the strengthening stress-strain relationship can be expressed by Eq. (5) [66]:

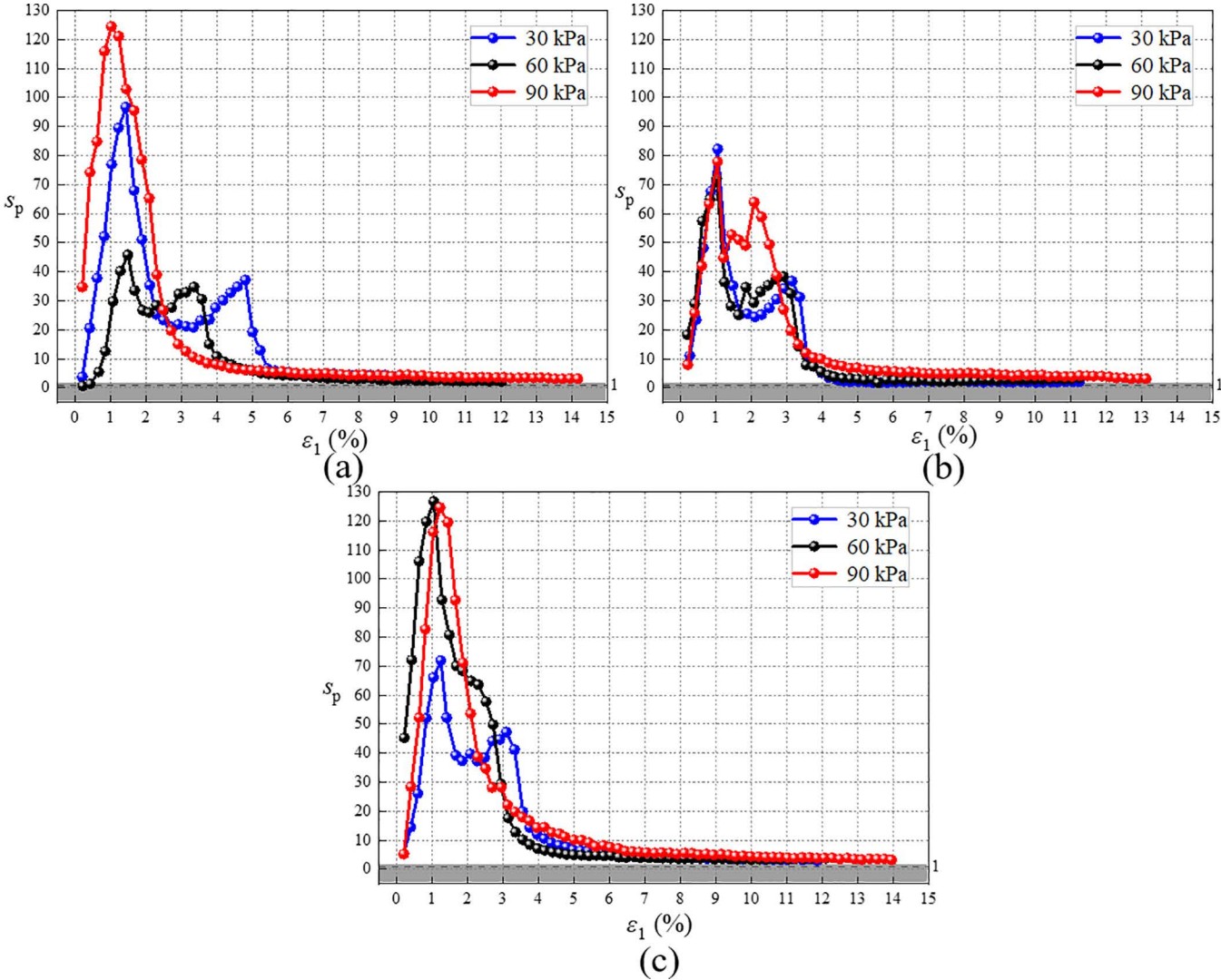

**Fig 15. The $s_p$-$\varepsilon_1$ curves of RMBF.** (a) 7d, (b) 14d, (c) 28d.

$$\begin{cases} (\sigma_1 - \sigma_3)_s = \dfrac{\varepsilon_1}{a + b \cdot \varepsilon_1} \\[3mm] \dfrac{\varepsilon_1}{(\sigma_1 - \sigma_3)_s} = a + b \cdot \varepsilon_1 \end{cases} \tag{5}$$

In Eq. (5), a and b are the experimental fitting constants of the RMBF triaxial compression test considering the strengthening parameter, which are obtained through the "ab line".

When calculating the material constants of the Duncan-Chang model, the stress-strain curve is assumed to fully conform to a hyperbolic shape. However, the stress-strain curve obtained from the triaxial compression test only approximates a hyperbolic shape and does not fully conform to it. There are numerous irregular data points. As shown in Fig 16, when drawing the "ab line," the test data points at both low and high stress levels frequently deviate from the "ab line." For this reason, Duncan et al. suggested using test data points within the 70% to 95% stress level range to determine the test parameters a and b, based on the analysis of extensive test data.

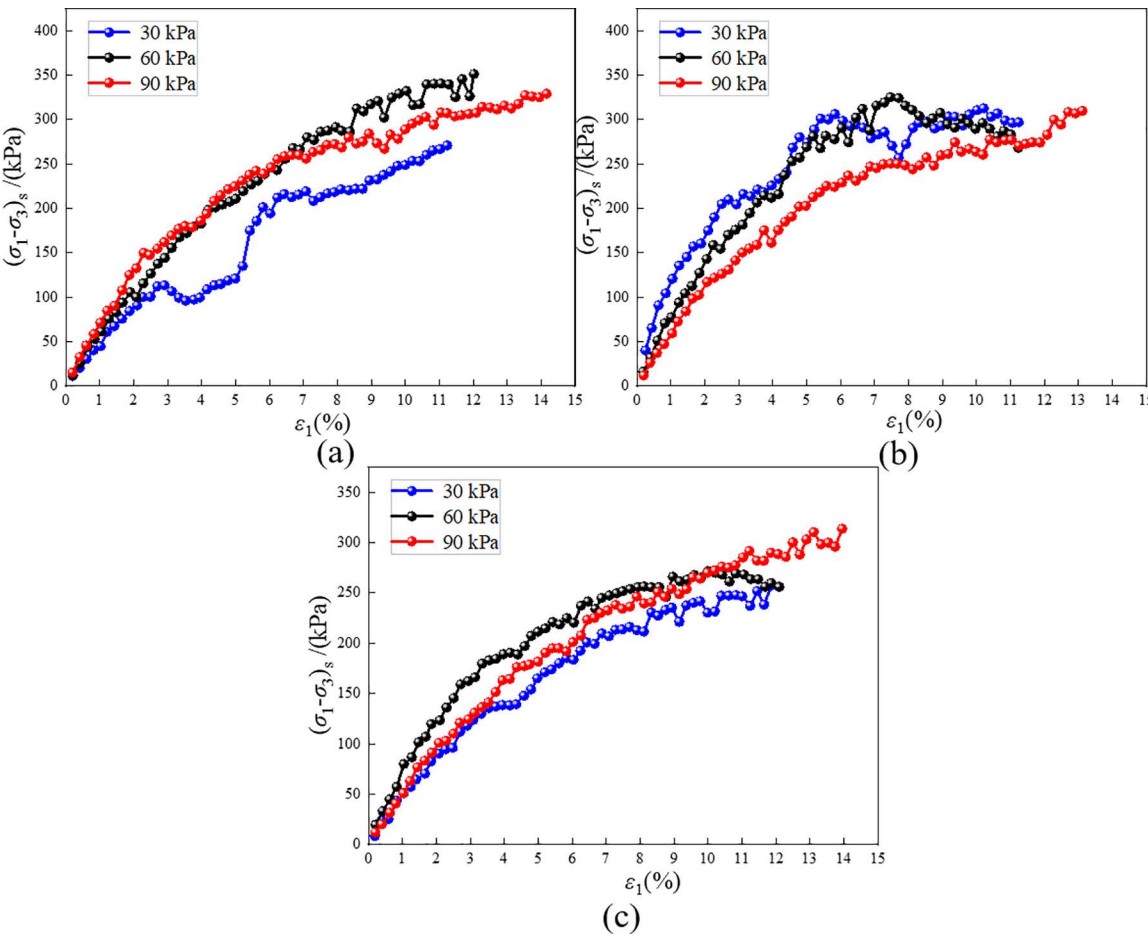

**Fig 16. Strengthening stress-strain curve of RMBF.** (a) 7d, (b) 14d, (c) 28d.

When plotting the $\varepsilon_1/(\sigma_1-\sigma_3)_s$ -$\varepsilon_1$ line of RMBF, it was observed that the "ab line" can be derived by selecting the test data point at the 40% to 50% stress level as the starting point. At this point, the strain at the starting point is approximately 5%. After adjusting the $\varepsilon_1/(\sigma_1-\sigma_3)_s$ -$\varepsilon_1$ curve of RMBF, a linear fit was performed to obtain the "ab line" shown in Fig 17. After processing, the Duncan-Chang test constants of RMBF, accounting for strengthening parameter at different ages and varying confining pressures, were determined (Table 7). It can be seen that the failure ratio $R_f$ of RMBF, considering strengthening parameter, ranges from 0.590 to 0.763, with significant variation across different ages, and it increases with age. Under 30 kPa, 60 kPa, and 90 kPa confining pressures, the damage ratio exhibits an increasing trend with rising confining pressure, which aligns with the conclusion in Section 3.4: Stress-strain curve that confining pressure significantly impacts the strength of RMBF. The average damage ratio of RMBF, accounting for strengthening parameter, is $R_f$=0.683.

The "$Kn$ line" illustrates the relationship between the initial tangent modulus and the confining pressure, as depicted in Fig 18. The intercept of the straight line on the ordinate is lg $K$, and the slope of the straight line is $n$. It calculated values are presented in Table 8. It can be seen that the material constants $K$ and $n$ in the Duncan-Chang model of RMBF, considering the strengthening parameter, exhibit a trend of alternating increases and decreases with age. The average values are: $K$=0.9571, $n$=0.3976.

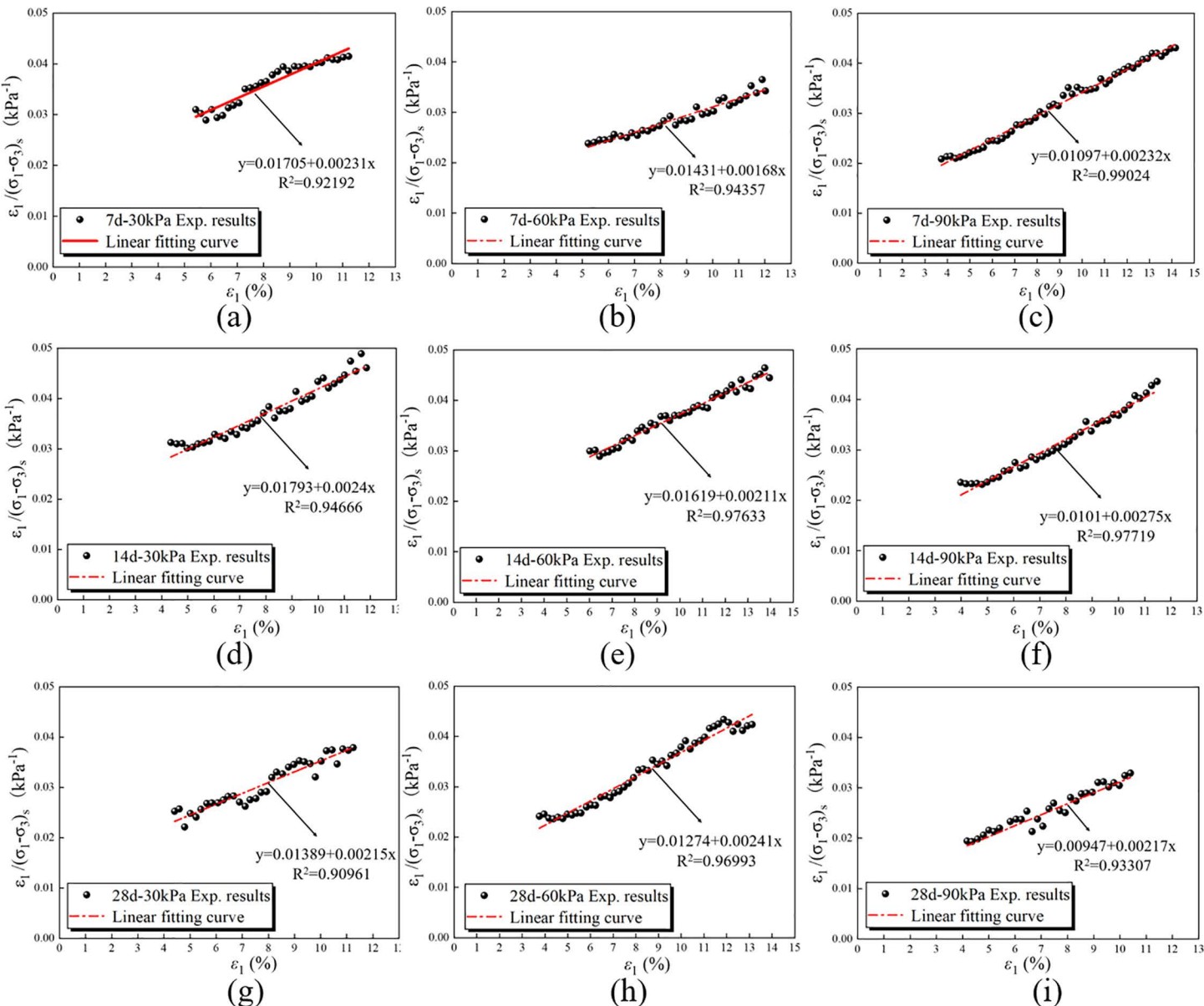

**Fig 17. The "ab line" of RMBF considering Sp.** (a) 7d-30kPa, (b) 7d-60kPa, (c) 7d-90kPa, (d) 14d-30kPa, (e) 14d-60kPa, (f) 14d-90kPa, (g) 28d-30kPa, (h) 28d-60kPa, (i) 28d-90kPa.

Based on Fig 16, the peak values of the strengthening stress $(\sigma_1-\sigma_3)_{sf}$ under different confining pressures can be determined. The Mohr circle is then drawn, and the cohesion and internal friction angle of the RMBF, considering the strengthening parameter, are calculated. The shear strength envelope at various ages is depicted in Fig 19. Cohesion increases with age, while the internal friction angle exhibits an opposite trend, decreasing as age increases (Table 9).

Fig 20 illustrates the "Df line". In the calculation, strain during the compaction stage is excluded, similar to the calculation of strengthening stress. Table 10 presents the data, revealing that the initial Poisson's ratio of RMBF decreases with age, while the material constant $D$ increases with age. Neither of the two shows a clear relationship with confining pressure.

**Table 7. Test constants $a$, $b$, $E_0$, $(\sigma_1-\sigma_3)_{\text{sult}}$ and $R_f$ of Duncan-Chang model for RMBF considering $S_p$.**

| Age/d | $\sigma_3$/kPa | $(\sigma_1-\sigma_3)_{st}$/kPa | $a$/kPa$^{-1}$ | $b$/kPa$^{-1}$ | $E_0$/kPa | $(\sigma_1-\sigma_3)_{\text{sult}}$/kPa | $R_f$ |
|---|---|---|---|---|---|---|---|
| 7d | 30 | 270.77 | 0.01705 | 0.00231 | 58.65 | 432.90 | 0.625 |
| | 60 | 351.12 | 0.01431 | 0.00168 | 69.88 | 595.24 | 0.590 |
| | 90 | 328.88 | 0.01097 | 0.00232 | 91.16 | 431.03 | 0.763 |
| 14d | 30 | 257.19 | 0.01793 | 0.00240 | 55.77 | 416.67 | 0.617 |
| | 60 | 269.91 | 0.01619 | 0.00211 | 61.77 | 473.93 | 0.570 |
| | 90 | 309.70 | 0.01010 | 0.00275 | 99.01 | 363.64 | 0.852 |
| 28d | 30 | 312.53 | 0.01389 | 0.00215 | 71.99 | 465.12 | 0.672 |
| | 60 | 325.34 | 0.01274 | 0.00241 | 78.49 | 414.94 | 0.784 |
| | 90 | 309.79 | 0.00947 | 0.00217 | 105.60 | 460.83 | 0.672 |

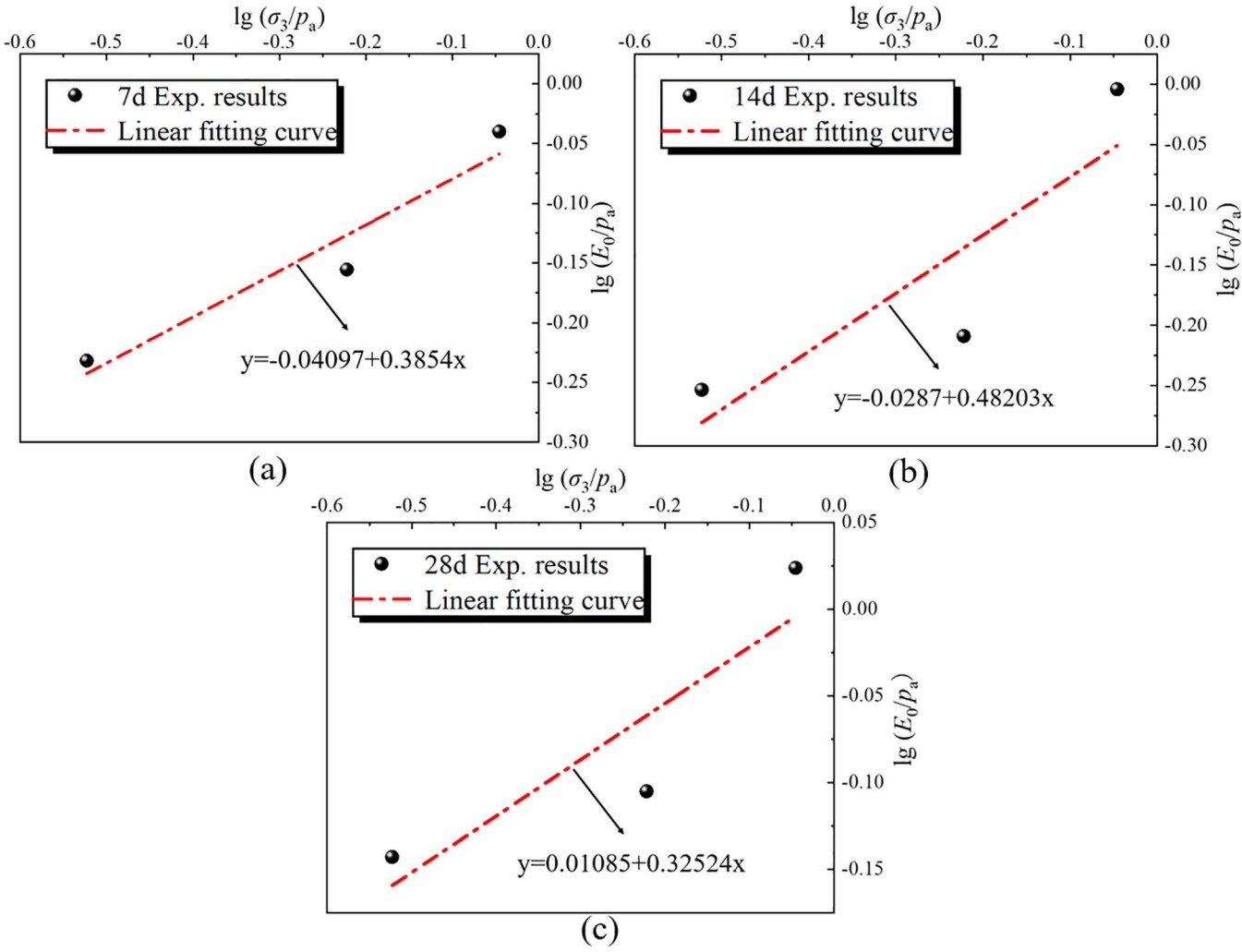

**Fig 18. The "$Kn$ line" of RMBF considering $Sp$.** (a) 7d, (b) 14d, (c) 28d.

**Table 8. Material constants $K$ and $n$ of Duncan-Chang model for RMBF considering $S_p$.**

| Age/d | lg $K$ | $K$ | $n$ |
|---|---|---|---|
| 7d | -0.04097 | 0.90998 | 0.38540 |
| 14d | -0.02870 | 0.93605 | 0.48203 |
| 28d | 0.01085 | 1.02530 | 0.32524 |
| Average value | | 0.9151 | 0.3976 |

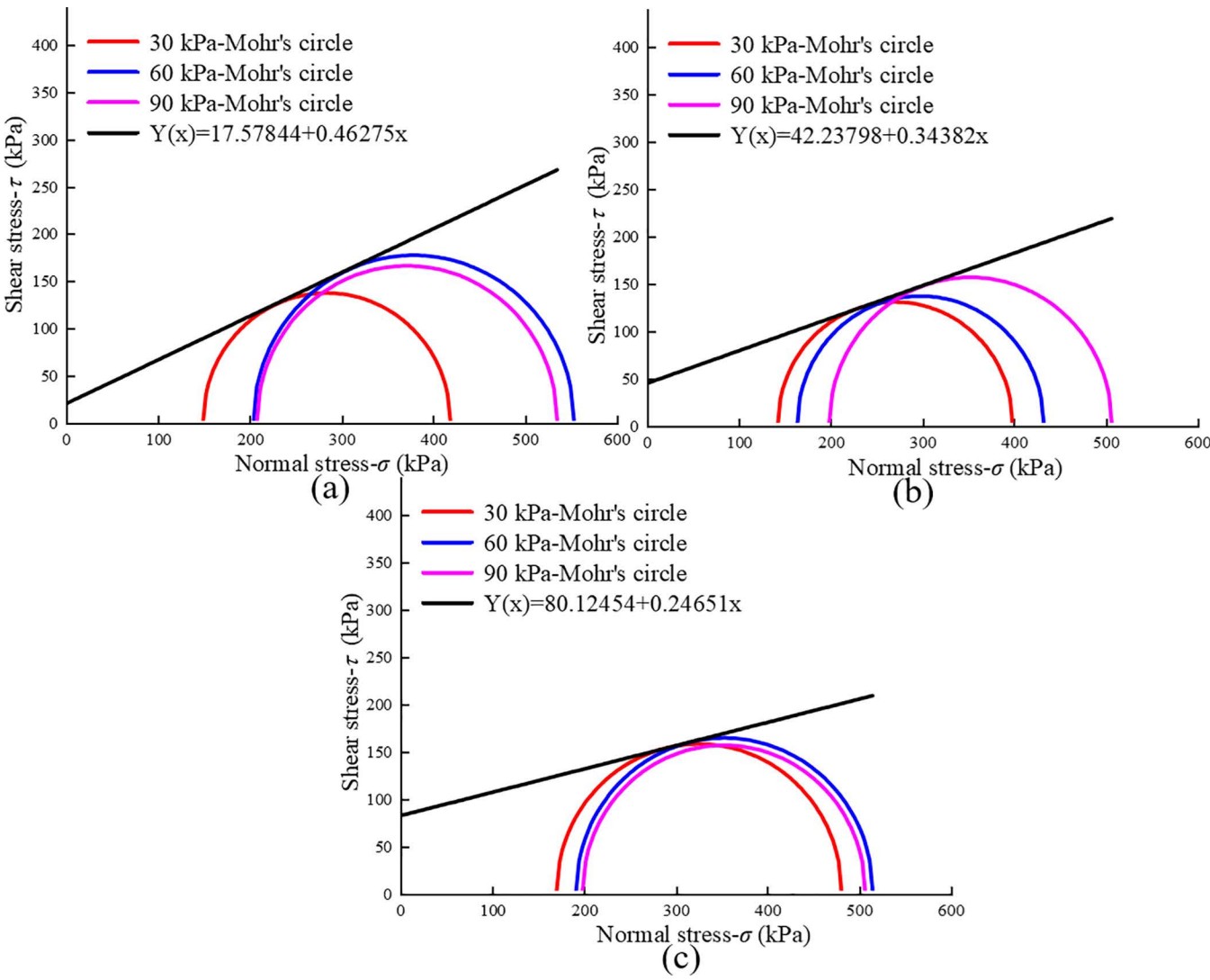

**Fig 19. Mohr stress circle of RMBF considering $Sp$.** (a) 7d, (b) 14d, (c) 28d.

As shown in Fig 21, unlike the "*GF* line" in the Duncan-Chang theory, this line exhibits an upward trend. This is because the confining pressure set in the test is less than atmospheric pressure, which results in an upward trend. The data is presented in Table 11. The material constants $G$ and $F$ in the RMBF Duncan-Chang model, considering the strengthening parameter, exhibit a decreasing trend with increasing age, with average values of $G = 0.5651$ and $F = 0.2580$.

Table 9. Material constants $c$ and $\varphi$ of Duncan-Chang model for RMBF considering $S_p$.

| Age/d | $c$/kPa | tan $\varphi$ | $\varphi$/° |
|---|---|---|---|
| 7d | 17.59 | 0.46 | 24.83 |
| 14d | 42.24 | 0.34 | 18.97 |
| 28d | 80.13 | 0.25 | 13.85 |
| Average value | 46.65 | | 19.22 |

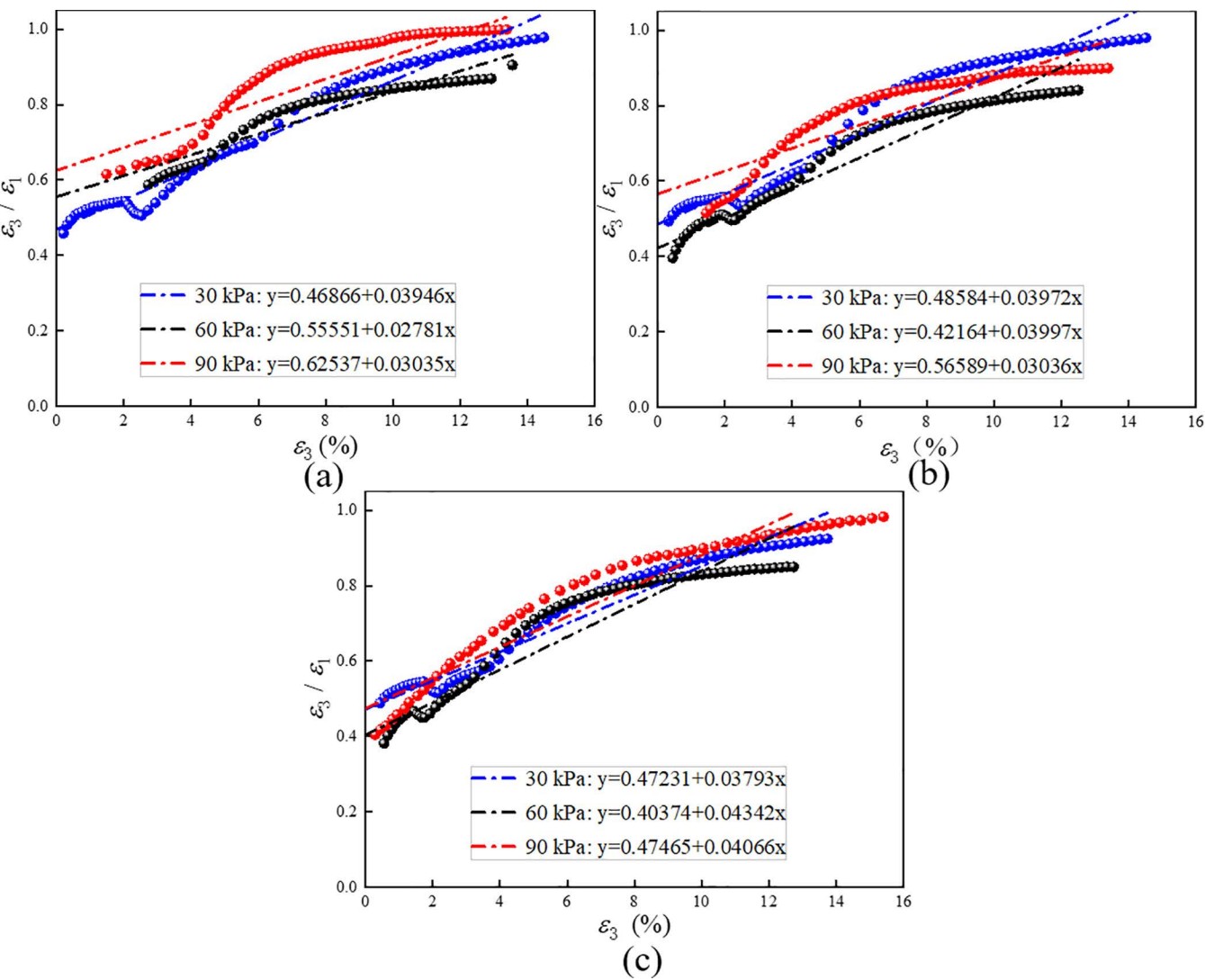

**Fig 20. The "Df line" of RMBF considering Sp.** (a) 7d, (b) 14d, (c) 28d.

As shown in Table 12, the eight material constants in the model all exhibit a relationship with age, and were fitted to produce Fig 22. It can be seen from the figure that, except for the material constant $n$, the other constants exhibit a linear relationship with age. We believe that when the slope of the fitting line is less than 0.005, the relationship between the material constant and age is negligible, allowing for the use of an average value. Thus, the material

**Table 10. Test constant $f$ and material constant $D$ of Duncan-Chang model for RMBF considering $S_p$.**

| Age/d | 7d | | | 14d | | | 28d | | |
|---|---|---|---|---|---|---|---|---|---|
| Confining pressure/kPa | 30 | 60 | 90 | 30 | 60 | 90 | 30 | 60 | 90 |
| $f(\mu_0)$ | 0.469 | 0.556 | 0.625 | 0.486 | 0.422 | 0.566 | 0.472 | 0.404 | 0.475 |
| Average value | | 0.550 | | | 0.491 | | | 0.450 | |
| $D$ | 0.039 | 0.028 | 0.030 | 0.040 | 0.040 | 0.030 | 0.038 | 0.043 | 0.041 |
| Average value | | 0.032 | | | 0.037 | | | 0.041 | |

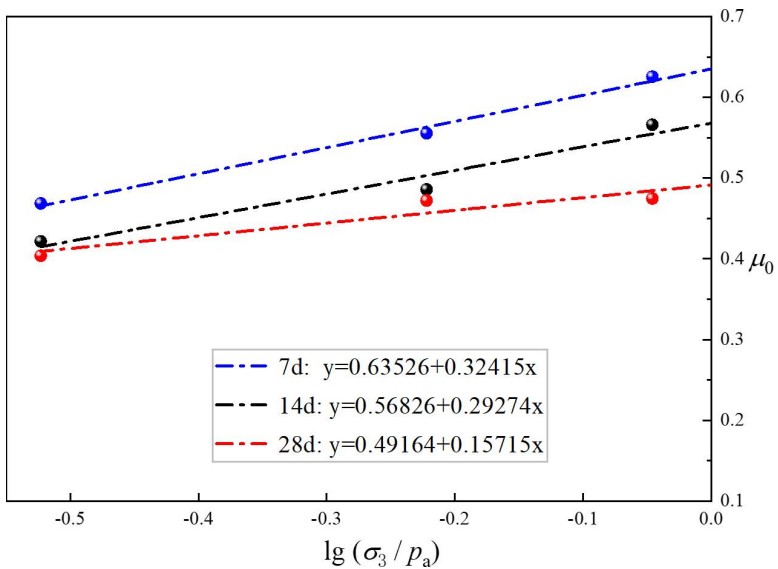

**Fig 21. The "$GF$ line" of RMBF.**

**Table 11. The material constants $G$ and $F$ of Duncan-Chang model for RMBF.**

| Age | $G$ | $F$ |
|---|---|---|
| 7d | 0.6353 | 0.3242 |
| 14d | 0.5683 | 0.2927 |
| 28d | 0.4916 | 0.1572 |
| Average value | 0.5651 | 0.2580 |

constants are $R_f = 0.683$ and $D = 0.037$, while the fitting relationships between the other six material constants and age are presented in Table 13.

## 4.4. Model validation

To verify the accuracy of the derived material constants for the Duncan-Chang model, the stress-strain relationship is constructed based on the aforementioned derivation results, allowing the stress-strain curve of RMBF to be reconstructed. The fitting relationship of the material constants from Table 13 is applied to the Duncan-Chang model calculation formula, deriving the strengthening stress-strain formula for RMBF, which incorporates the $S_p$. The calculation process is presented in Eq. (6).

**Table 12. List of material constants of Duncan-Chang model for RMBF considering $S_p$.**

| Age/d | K | n | $R_f$ | c/kPa | φ/° | D | G | F |
|---|---|---|---|---|---|---|---|---|
| 7d | 0.910 | 0.385 | 0.659 | 17.59 | 24.83 | 0.032 | 0.635 | 0.324 |
| 14d | 0.936 | 0.482 | 0.679 | 42.24 | 18.97 | 0.037 | 0.568 | 0.293 |
| 28d | 1.025 | 0.325 | 0.709 | 80.13 | 13.85 | 0.041 | 0.492 | 0.157 |
| Average value | 0.915 | 0.398 | 0.683 | 46.65 | 19.22 | 0.037 | 0.565 | 0.258 |

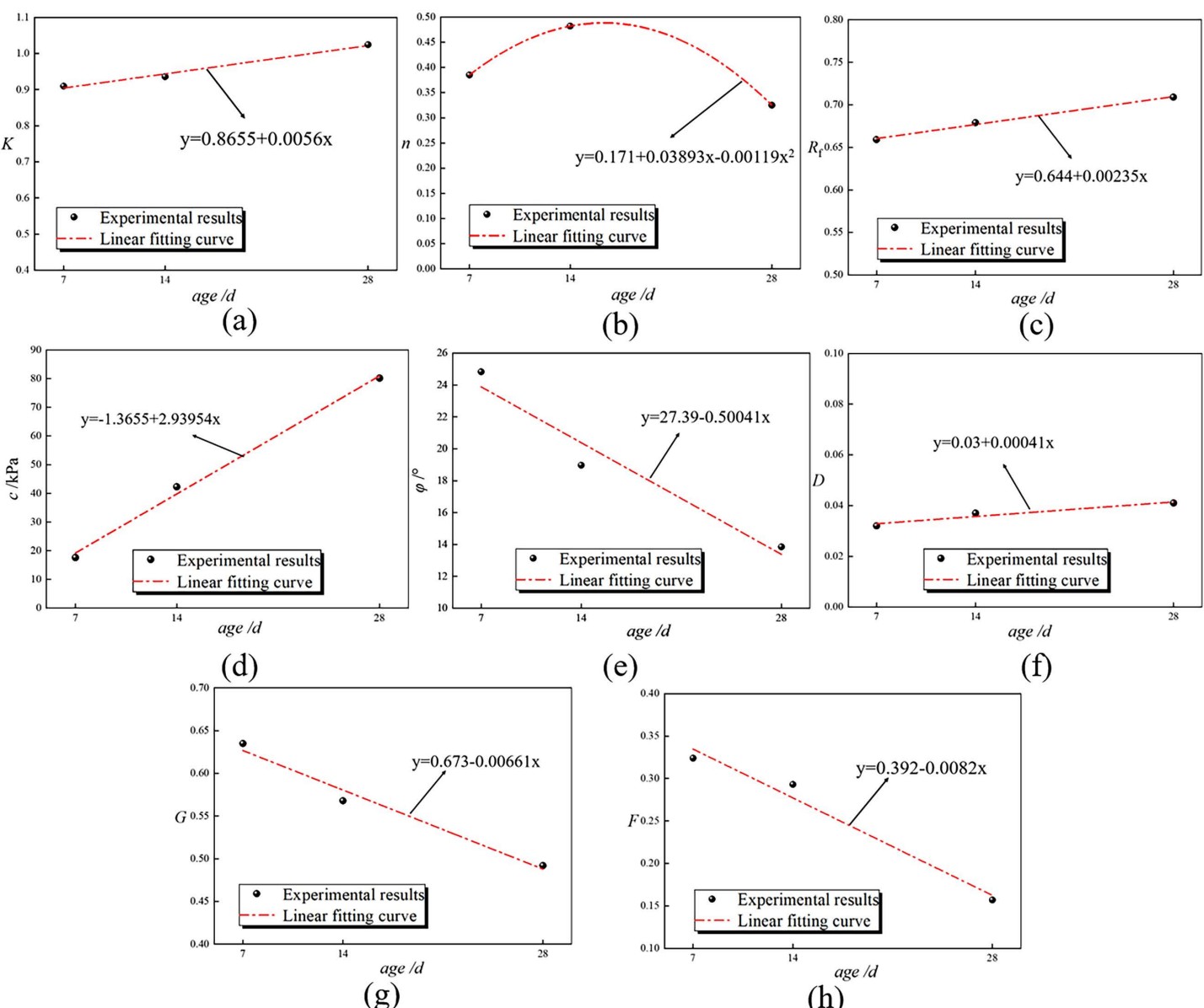

**Fig 22. Fitting lines for material constants of Duncan-Chang model for RMBF considering Sp.** (a) K-age, (b) n-age, (c) $R_f$-age, (d) c-age, (e) φ-age, (f) D-age, (g) G-age, (h) F-age.

**Table 13. List of fitting equations for material constants.**

| Material constants | Fitting relationship ($t$-age, d) | Function coefficient value |
|---|---|---|
| $K$ | $K = K_0 + K_s \cdot t$ | $K_0 = 0.8655$, $K_s = 0.0056$ |
| $n$ | $n = n_0 + n_s \cdot t + n_c \cdot t^2$ | $n_0 = 0.1710$, $n_s = 0.0389$, $n_c = -0.0012$ |
| $R_f$ | $R_f = 0.683$ | \ |
| $c$ | $c = c_0 + c_s \cdot t$ | $c_0 = -1.3655$, $c_s = 2.9395$ |
| $\varphi$ | $\varphi = \varphi_0 + \varphi_s \cdot t$ | $\varphi_0 = 27.39$, $\varphi_s = 0.5004$ |
| $D$ | $D = 0.037$ | \ |
| $G$ | $G = G_0 + G_s \cdot t$ | $G_0 = 0.6730$, $G_s = 0.0066$ |
| $F$ | $F = F_0 + F_s \cdot t$ | $F_0 = 0.3920$, $F_s = 0.0082$ |

$$(\sigma_1 - \sigma_3)_s = \frac{\varepsilon_1}{a + b \cdot \varepsilon_1} = \frac{\varepsilon_1}{\dfrac{1}{E_0} + \dfrac{R_f}{(\sigma_1 - \sigma_3)_f} \cdot \varepsilon_1} = \frac{\varepsilon_1}{K^{-1} \cdot p_a^{-1} \cdot \left(\dfrac{\sigma_3}{p_a}\right)^{-n} + \dfrac{R_f \cdot (1 - \sin\varphi)}{2 \cdot c \cdot \cos\varphi + 2 \cdot \sigma_3 \cdot \sin\varphi} \cdot \varepsilon_1}$$

(6)

$$= \frac{\varepsilon_1}{(K_0 + K_s \cdot t)^{-1} \cdot p_a^{-1} \cdot \left(\dfrac{\sigma_3}{p_a}\right)^{-(n_0 + n_s \cdot t + n_c \cdot t^2)} + \dfrac{R_f \cdot [1 - \sin(\varphi_0 + \varphi_s \cdot t)]}{2 \cdot (c_0 + c_s \cdot t) \cdot \cos(\varphi_0 + \varphi_s \cdot t) + 2 \cdot \sigma_3 \cdot \sin(\varphi_0 + \varphi_s \cdot t)} \cdot \varepsilon_1}$$

In Eq. (6), the values of $K_0$, $K_s$, $p_a$, $n_0$, $n_s$, $n_c$, $R_f$, $c_0$, $c_s$, $\varphi_0$, and $\varphi_s$ are constants, while the remaining variables are $(\sigma_1 - \sigma_3)_s$, $\varepsilon_1$, $\sigma_3$, and $t$, where $\sigma_3$ and $t$ represent the test conditions of the triaxial compression test, specifically the confining pressure and age, and $(\sigma_1 - \sigma_3)_s$ and $\varepsilon_1$ constitute the outcomes of the triaxial compression test, specifically the strengthening stress-strain curve.

The strengthening stress obtained from theoretical calculations is transformed into RMBF stress using Eq. (4), and the comparison with the experimental results is shown in Fig 23. The peak stress of the theoretical curve is slightly higher than the test value, with a difference of approximately 10%. Additionally, the fluctuation amplitude of the curve in the plastic deformation stage also increases. This occurs because the peak value appears in the small strain region. When performing theoretical calculations, the strengthening stress in the small strain region is higher than the test value. Therefore, when transformed using Eq. (4), the stress is naturally higher than the test value. However, the trend of the theoretical curve is highly consistent with that of the test curve, indicating that the theoretical calculation can accurately reflect the test value. The Duncan-Chang model of RMBF, which considers the $S_p$ introduced in this paper, can effectively predict the triaxial compression behavior of RMBF at various ages and confining pressures, demonstrating the model's feasibility.

The constitutive model presented in this study proves to be highly valuable for the design of future tunnel inverts. It effectively forecasts the behavior of RMBF under various environmental conditions, offering essential insights for maintaining structural integrity. Nevertheless, certain limitations of the model must be acknowledged. For instance, the model presumes that RMBF behaves as a uniform, continuous material, an assumption that may not always hold true in natural settings. Furthermore, it does not take into account groundwater influences, which could cause discrepancies in the predicted performance. To ensure the model's successful application, engineers should account for these limitations and modify the model parameters accordingly when implementing it in real-world projects.

While the model's validation against experimental data demonstrates its reliability, several factors could introduce errors and impact the accuracy of the results. Variability in sample

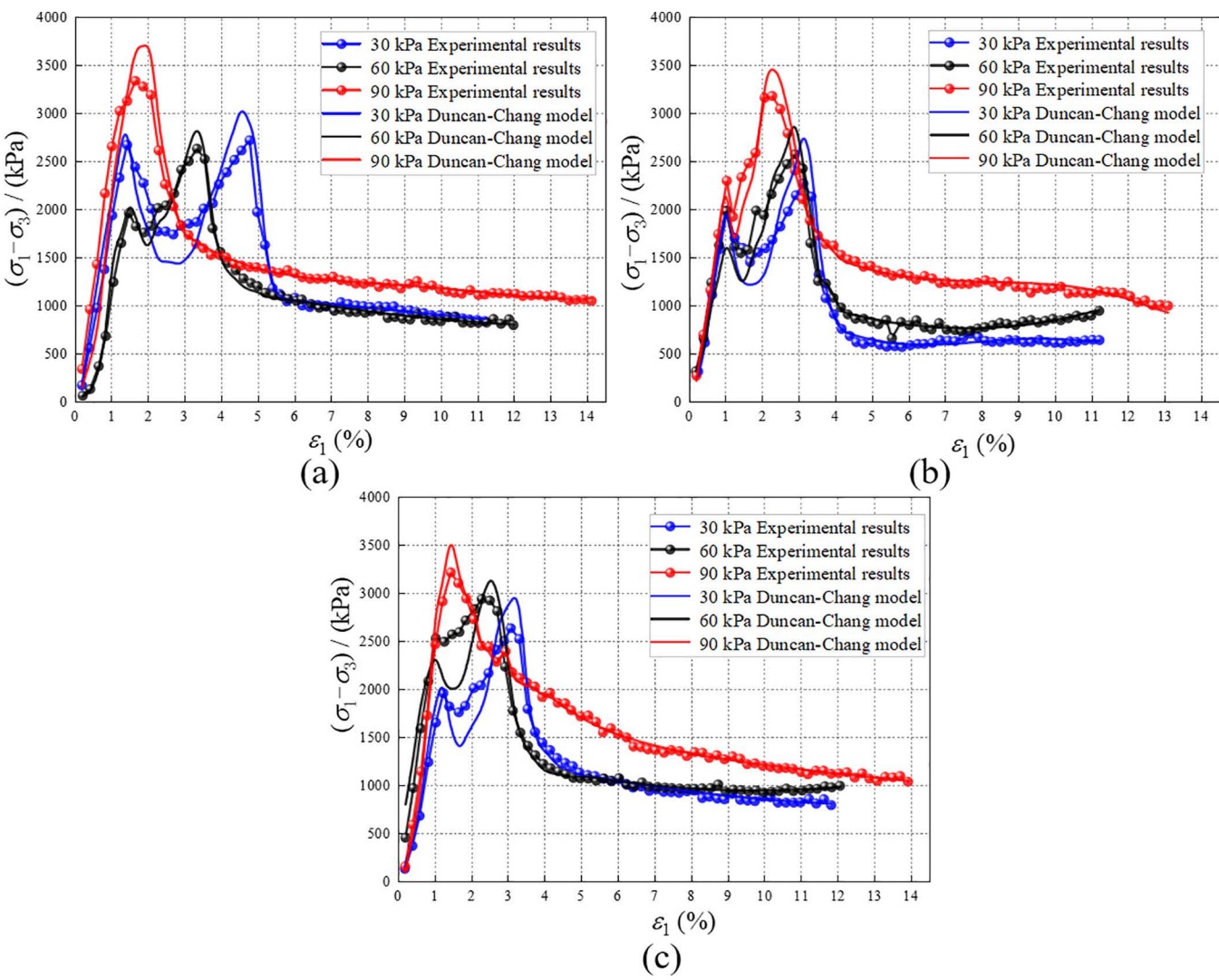

**Fig 23. The ($\sigma_1$-$\sigma_3$)-$\varepsilon_1$ curves calculated by Duncan-Chang model for RMBF considering *Sp*.** (a) 7d, (b) 14d, (c) 28d.

preparation—such as achieving consistent moisture content and ensuring uniform sample density—can result in differences between test samples. Additionally, slight variations in data may arise from the precision limits of the triaxial testing apparatus in regulating the confining pressure. Although efforts are made to minimize these factors, it is essential to acknowledge their potential impact when extrapolating from laboratory conditions to field applications.

## 5. Conclusions

This paper identifies the optimal ratio of RMBF for highway tunnel invert filling through experiments and proposes the $S_p$ of RMBF based on triaxial compression test results, the definition of soil structural parameters, and Duncan-Chang theory, subsequently establishing the constitutive model of RMBF. Finally, the experimental data are compared and analyzed against the results of theoretical calculations to verify the validity of the constitutive model.

(1) The components and mass ratios of RMBF were determined as follows: red mud 51.7%, cement 5.8%, steel slag 11.6%, slaked lime 5.8%, EDTA-2Na 5.8%, water 16.1%, and

TSD curing agent 3.2%. The unconfined compressive strength of the 28-day RMBF was 7.37 MPa, meeting the required strength criteria. When used for highway tunnel invert filling, the compaction degree must reach 96%. One cubic meter of RMBF can recycle 1.068 tons of red mud.

(2) The stress-strain curve of the RMBF exhibits a distinct peak, indicating strain-softening characteristics. The deformation and failure process under triaxial compression can be divided into five stages: compaction, elastic deformation, plastic deformation, yield, and post-peak stability. As the confining pressure increases, both the elastic modulus and stress peak of the RMBF increase, indicating that confining pressure significantly influences the strength of the RMBF.

(3) The strengthening parameter of RMBF were proposed and defined. The strain softening curve of RMBF was adjusted to a strain hardening curve using $S_p$, after which the constitutive model of RMBF, incorporating the strengthening parameter, was derived. The eight material constants of the Duncan-Chang model were determined, and the theoretical stress-strain curves of RMBF at different ages and confining pressures were obtained through fitting calculations. The theoretical curves were highly consistent with the test curves, with an error of approximately 10%.

In conclusion, the findings of this study on the mechanical properties of RMBF and the corresponding structural constitutive model offer valuable solutions to engineering challenges related to its use in tunnel construction. These results also lay the groundwork for future engineering projects involving RMBF. However, additional research is required to explore the model's behavior under varying environmental conditions, including extreme seasonal fluctuations, which could influence the durability of RMBF. Moreover, experimental investigations into the modification of red mud with additives like stabilizers can further enhance the model's accuracy, contributing to more reliable engineering applications.

## Supporting information

**S1 Table. Red mud emissions and comprehensive utilization rate in China in the past decade** (Fig 1)**.** Data of red mud emissions and comprehensive utilization rate.
(DOCX)

**S2 Table. Compaction test curves** (Fig 5)**.** Test results of compaction test.
(DOCX)

**S3 Table. Unconfined compressive strength of MRM** (Fig 7)**.** Test results of unconfined compressive strength.
(DOCX)

**S4 Table. The changing trend of unconfined compressive strength of MRM** (Fig 8)**.** Data of the changing trend of unconfined compressive strength.
(DOCX)

**S5 Table. Unconfined compressive strength of MRM after adding soil stabilizers** (Fig 9)**.** Test results of unconfined compressive strength of MRM after adding soil stabilizers.
(DOCX)

**S6 Table. The (σ1-σ3)-ε1 curves of RMBF with confining pressure under different ages** (Fig 10)**.** Test results of $(\sigma_1-\sigma_3)$-$\varepsilon_1$ curves of RMBF with confining pressure under different ages.
(DOCX)

**S7 Table. The changing trend of E and (σ1-σ3)f of RMBF** (Fig 12). Test results of The changing trend of $E$ and $(\sigma_1-\sigma_3)_f$ of RMBF.
(DOCX)

**S8 Table. The (σ1-σ3)-ε1 curves of URM with confining pressure under different ages** (Fig 13). Test curve and fitting curve results.
(DOCX)

**S9 Table. The (σ1-σ3)-ε1 curves of MRM with confining pressure under different ages** (Fig 14). Test curve and fitting curve results.
(DOCX)

**S10 Table. The sp-ε1 curves of RMBF** (Fig 15). Data of $s_p$-$\varepsilon_1$ curves.
(DOCX)

**S11 Table. Strengthening stress-strain curve of RMBF** (Fig 16). Data of strengthening stress-strain curve.
(DOCX)

**S12 Table. The "ab line" of RMBF considering Sp** (Fig 17). Data of "ab line" of RMBF considering Sp.
(DOCX)

**S13 Table. The "Kn line" of RMBF considering Sp** (Fig 18). Data of " Kn line" of RMBF considering Sp.
(DOCX)

**S14 Table. Mohr stress circle of RMBF considering Sp** (Fig 19). Data of mohr stress circle of RMBF considering Sp.
(DOCX)

**S15 Table. The "Df line" of RMBF considering Sp** (Fig 20). Data of "Df line" of RMBF considering Sp.
(DOCX)

**S16 Table. The "GF line" of RMBF** (Fig 21). Data of "GF line" of RMBF.
(DOCX)

**S17 Table. Fitting lines for material constants of Duncan-Chang model for RMBF considering Sp** (Fig 22). Data of material constants of Duncan-Chang model for.
(DOCX)

**S18 Table. The (σ1-σ3)-ε1 curves calculated by Duncan-Chang model for RMBF considering Sp** (Fig 23). Data of (σ1-σ3)-ε1 curves calculated by Duncan-Chang model for RMBF considering Sp.
(DOCX)

## Author contributions

**Conceptualization:** Junying Rao, Hongchao Cui.

**Data curation:** Xiaolong Song.

**Methodology:** Junying Rao.

**Resources:** Junying Rao, Mingwei He.

**Software:** Hongchao Cui.

**Supervision:** Junying Rao, Mingwei He.

**Visualization:** Hongchao Cui.

**Writing – original draft:** Hongchao Cui.

**Writing – review & editing:** Xiaolong Song.

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
