## [Decision Letter · Decision Letter 0]

23 Feb 2025

PONE-D-25-03386Material composition and constitutive model development of red mud-based filler for highway tunnel invert filling applications: A comprehensive studyPLOS ONE

Dear Dr. Rao, Thank you for submitting your manuscript to PLOS ONE. After careful consideration, we feel that it has merit but does not fully meet PLOS ONE’s publication criteria as it currently stands. Therefore, we invite you to submit a revised version of the manuscript that addresses the points raised during the review process.==============================

We look forward to receiving your revised manuscript.

Kind regards,

Fahd Saeed Alakbari, Ph.D.

Academic Editor

PLOS ONE

Journal Requirements:

Reviewers' comments:

Reviewer's Responses to Questions

**Comments to the Author**

1. Is the manuscript technically sound, and do the data support the conclusions?

Reviewer #1: Yes

2. Has the statistical analysis been performed appropriately and rigorously? 

Reviewer #1: Yes

3. Have the authors made all data underlying the findings in their manuscript fully available?

Reviewer #1: Yes

4. Is the manuscript presented in an intelligible fashion and written in standard English?

Reviewer #1: Yes

5. Review Comments to the Author

Reviewer #1: comments given in atthed file

Let's give the sample specifications in a small table

Are the layer thicknesses different from the values in Figure 3? Explain...

Could it be that the rates of increase are mutually confused???

The explanation of the figure is prone to misunderstanding in this way, at least it will be better to understand if the values determined by the ratio are shown on the graph

Can you give the desired min max values in Code 23 in a table?

give the correlation coefficients to the curve fitting equations....

may be given regression corr coef for the models ?

Can you explain what effect groundwater will have on the RBMF fill in the tunnel?

Can we make an estimate of the RMBF recycling tonnes in tunnel design per km?

6. PLOS authors have the option to publish the peer review history of their article (what does this mean? ). If published, this will include your full peer review and any attached files.

**Do you want your identity to be public for this peer review?** For information about this choice, including consent withdrawal, please see our Privacy Policy .

Reviewer #1: **Yes: ** Dr. Mesut TIĞDEMİR

---

## [Author Response · Author response to Decision Letter 0]

26 Feb 2025

Dear editors and reviewers,

We sincerely appreciate your letter and the reviewers' comments regarding our manuscript entitled "Material composition and constitutive model development of red mud-based filler for highway tunnel invert filling applications: A comprehensive study" (Manuscript Number: PONE-D-25-03386). These comments are invaluable and extremely helpful in revising and improving our manuscript, and they provide significant guidance for our research. We have carefully reviewed the comments and made corrections, which we hope will meet your approval.

The editor's and reviewers' comments are outlined below in italicized font, with specific concerns numbered. Our responses are provided in blue text, and the changes made to the manuscript are highlighted in yellow text. Specific changes in the responses are in bold text.

The following is the original email:

PONE-D-25-03386

Material composition and constitutive model development of red mud-based filler for highway tunnel invert filling applications: A comprehensive study

PLOS ONE

Dear Dr. Rao,

Thank you for submitting your manuscript to PLOS ONE. After careful consideration, we feel that it has merit but does not fully meet PLOS ONE’s publication criteria as it currently stands. Therefore, we invite you to submit a revised version of the manuscript that addresses the points raised during the review process.

We look forward to receiving your revised manuscript.

Kind regards,

Fahd Saeed Alakbari, Ph.D.

Academic Editor

PLOS ONE

Journal Requirements:

Reviewers' comments:

Reviewer's Responses to Questions

Comments to the Author

1. Is the manuscript technically sound, and do the data support the conclusions?

Reviewer #1: Yes

2. Has the statistical analysis been performed appropriately and rigorously?

Reviewer #1: Yes

3. Have the authors made all data underlying the findings in their manuscript fully available?

Reviewer #1: Yes

4. Is the manuscript presented in an intelligible fashion and written in standard English?

Reviewer #1: Yes

5. Review Comments to the Author

Reviewer #1: comments given in atthed file

Let's give the sample specifications in a small table

Are the layer thicknesses different from the values in Figure 3? Explain...

Could it be that the rates of increase are mutually confused???

The explanation of the figure is prone to misunderstanding in this way, at least it will be better to understand if the values determined by the ratio are shown on the graph

Can you give the desired min max values in Code 23 in a table?

give the correlation coefficients to the curve fitting equations....

may be given regression corr coef for the models ?

Can you explain what effect groundwater will have on the RBMF fill in the tunnel?

Can we make an estimate of the RMBF recycling tonnes in tunnel design per km?

6. PLOS authors have the option to publish the peer review history of their article (what does this mean?). If published, this will include your full peer review and any attached files.

Do you want your identity to be public for this peer review? For information about this choice, including consent withdrawal, please see our Privacy Policy.

Reviewer #1: Yes: Dr. Mesut TIĞDEMİR

While revising your submission, please upload your figure files to the Preflight Analysis and Conversion Engine (PACE) digital diagnostic tool, https://pacev2.apexcovantage.com/. PACE helps ensure that figures meet PLOS requirements. To use PACE, you must first register as a user. Registration is free. Then, login and navigate to the UPLOAD tab, where you will find detailed instructions on how to use the tool. If you encounter any issues or have any questions when using PACE, please email PLOS at figures@plos.org. Please note that Supporting Information files do not need this step. 

Journal Requirements:

Reply:

Thank you for your valuable comment regarding the manuscript formatting. We have reviewed and revised the manuscript to fully comply with PLOS ONE's formatting requirements. Please see the revised manuscript.

Reply:

Thank you for your comment regarding the sharing of author-generated code. We would like to clarify that our manuscript does not include any author-generated code underpinning the findings. The only reference to code in the manuscript pertains to the serial number field used for inserting references via a reference manager. As no code is involved in our work, we trust that this requirement does not apply to our submission.

At the same time, we think that the word "code" may cause ambiguity. After discussion, we decided to replace "code" with "specification", which can more accurately express the meaning of Chinese standards. Please see pages 6 and 9-10 of the revised manuscript.

The specific modifications are as follows:

“The compaction test was conducted following the Highway Geotechnical Test Specification (JTG 3430-2020) [60] to determine the maximum dry density and the optimum moisture content of the mixture. The mixture was prepared by the dry soil method, beginning with a moisture content of 18%, increasing by 2% increments. The mixture was thoroughly mixed and then tested after being sealed for 24 hours. The experiment consisted of 7 groups of tests, with a total of 35 specimens (Table 2). The test utilized the heavy II-2 compaction method, involving compaction in three layers.”

“The UCS test was conducted following the Test Specification for Stabilized Materials with Inorganic Binders for Highway Engineering (JTG E51-2009) [61]. The specimens were prepared based on the optimum moisture content. To account for compaction effects during construction, three compaction degrees of 90%, 93%, and 96% were selected. The specimens were formed using hydraulic molds and placed in a standard curing box (temperature 20±2°C, humidity above 95%) for 7 days and 28 days. Three specimens were prepared in each group, resulting in a total of 126 specimens (Table 3). The test instrument used was a YYW-11 unconfined pressure instrument, manufactured by Tianjin Gangyuan Testing Instrument Factory, with a loading rate of 3 mm/min.”

“At a composite modified material content of 30% and a compaction degree of 96%, the 28d unconfined compressive strength of MRM reaches 6.22 MPa, which is still lower than the 7.2 MPa compressive strength requirement for invert filling in the specification [20]. To address this, soil stabilizers were introduced, with resin polymer as the main component. In this study, two types of stabilizers, Tushengda (TSD) and Suzhou Luxing (SZLX), were used to mix with MRM containing 30% composite modified materials. As shown in Fig. 9, with the addition of 5-fold diluted TSD stabilizer, the 28d unconfined compressive strength of MRM reaches 7.37 MPa, which meets the requirements of the specification.”

Reply:

Thank you for your comments on our fieldwork permit. We would like to clarify that material acquisition for this study was conducted at a red mud dump site in Qingzhen, Guiyang, Guizhou Province, and no special permit was required to enter the site. And no specific authorization is required to collect samples from the site.

It should be further explained that, because Editor Maidelyn R. Peregrin believed that Figure 4 might involve copyright issues during the technical review, although these two photos were taken by us personally on site, we decided to delete Fig.4 to avoid unnecessary disputes.

Reply:

Thank you for pointing out the inconsistency between the ‘Funding Information’ and ‘Financial Disclosure’ sections of our manuscript. We apologize for the oversight, which occurred when we were filling out the relevant content in the submission system. We will make the necessary corrections to ensure that both sections are consistent and accurately reflect the funding sources. The revised manuscript will include the corrected information when resubmitted.

Reply:

Thank you for your comment regarding the reference list. We have carefully reviewed all the references cited in the manuscript to ensure that they are complete and accurate. In addition to making the necessary changes, we have also verified the status of each cited article to confirm that none of them have been retracted.

We can confirm that all ci

---

## [Decision Letter · Decision Letter 1]

14 Mar 2025

Material composition and constitutive model development of red mud-based filler for highway tunnel invert filling applications: A comprehensive study

PONE-D-25-03386R1

Dear Dr. Junying Rao,

We’re pleased to inform you that your manuscript has been judged scientifically suitable for publication and will be formally accepted for publication once it meets all outstanding technical requirements.

Kind regards,

Fahd Saeed Alakbari, Ph.D.

Academic Editor

PLOS ONE

Additional Editor Comments (optional):

Reviewers' comments:

Reviewer's Responses to Questions

**Comments to the Author**

1. If the authors have adequately addressed your comments raised in a previous round of review and you feel that this manuscript is now acceptable for publication, you may indicate that here to bypass the “Comments to the Author” section, enter your conflict of interest statement in the “Confidential to Editor” section, and submit your "Accept" recommendation.

Reviewer #1: All comments have been addressed

2. Is the manuscript technically sound, and do the data support the conclusions?

Reviewer #1: (No Response)

3. Has the statistical analysis been performed appropriately and rigorously? 

Reviewer #1: (No Response)

4. Have the authors made all data underlying the findings in their manuscript fully available?

Reviewer #1: (No Response)

5. Is the manuscript presented in an intelligible fashion and written in standard English?

Reviewer #1: (No Response)

6. Review Comments to the Author

Reviewer #1: (No Response)

7. PLOS authors have the option to publish the peer review history of their article (what does this mean? ). If published, this will include your full peer review and any attached files.

**Do you want your identity to be public for this peer review?** For information about this choice, including consent withdrawal, please see our Privacy Policy .

Reviewer #1: **Yes: ** Mesut TIĞDEMİR

---

## [Editor Report · Acceptance letter]

PONE-D-25-03386R1

PLOS ONE

Dear Dr. Rao,

I'm pleased to inform you that your manuscript has been deemed suitable for publication in PLOS ONE. Congratulations! Your manuscript is now being handed over to our production team.

Kind regards,

on behalf of

Dr. Fahd Saeed Alakbari

Academic Editor

PLOS ONE